# Gravity waves generated by the Hunga Tonga-Hunga Ha'apai volcanic eruption and their global propagation in the mesosphere/lower thermosphere observed by meteor radars and modeled with the High-Altitude General Mechanistic Circulation Model

Gunter Stober[1], Sharon L. Vadas[2], Erich Becker[2], Alan Liu[3], Alexander Kozlovsky[4], Diego Janches[5], Zishun Qiao[3], Witali Krochin[1], Guochun Shi[1], Wen Yi[6], Jie Zeng[6,1], Peter Brown[7,8], Denis Vida[7], Neil Hindley[24], Christoph Jacobi[9], Damian Murphy[10], Ricardo Buriti[11], Vania Andrioli[12,13], Paulo Batista[12], John Marino[14], Scott Palo[14], Denise Thorsen[15], Masaki Tsutsumi[16,17], Njål Gulbrandsen[18], Satonori Nozawa[19], Mark Lester[20], Kathrin Baumgarten[21], Johan Kero[22], Evgenia Belova[22], Nicholas Mitchell[23,24], Tracy Moffat-Griffin[23], and Na Li[25]

[1]Institute of Applied Physics & Oeschger Center for Climate Change Research, Microwave Physics, University of Bern, Bern, Switzerland
[2]North West Research Associates (NWRA), Boulder, Colorado, USA
[3]Center for Space and Atmospheric Research and Department of Physical Sciences, Embry-Riddle Aeronautical University, Daytona Beach, Florida, USA
[4]Sodankylä Geophysical Observatory, University of Oulu, Finland
[5]ITM Physics Laboratory, NASA Goddard Space Flight Center, Greenbelt, MD, USA
[6]CAS Key Laboratory of Geospace Environment/CAS Center for Excellence in Comparative Planetology, Anhui Mengcheng Geophysics National Observation and Research Station, University of Science and Technology of China, Hefei, China
[7]Dept. of Physics and Astronomy, University of Western Ontario, London, Ontario, Canada
[8]Western Institute for Earth and Space Exploration, University of Western Ontario, London, Ontario, Canada
[9]Institute for Meteorology, Leipzig University, Leipzig, Germany
[10]Australian Antarctic Division, Kingston, Tasmania, Australia
[11]Department of Physics, Federal University of Campina Grande, Campina Grande, Brazil
[12]National Institute for Space Research (INPE), São José dos Campos, Brazil
[13]China-Brazil Joint Laboratory for Space Weather, NSSC/INPE, São José dos Campos, Brazil
[14]Colorado Center for Astrodynamics Research, University of Colorado Boulder, Boulder, CO, USA
[15]University of Alaska, Fairbanks, USA
[16]National Institute of Polar Research, Tachikawa, Japan
[17]The Graduate University for Advanced Studies (SOKENDAI), Tokyo, Japan
[18]Tromsø Geophysical Observatory, UiT - The Arctic University of Norway, Tromsø, Norway
[19]Institute for Space-Earth Environmental Research, Nagoya University, Japan
[20]University of Leicester, Leicester, UK
[21]Fraunhofer Institute for Ceramic Technologies and Systems IKTS, Smart Ocean Technologies, Rostock, Germany
[22]Swedish Institute of Space Physics (IRF), Kiruna, Sweden
[23]British Antarctic Survey, Cambridge, CB3 0ET, UK
[24]Department of Electronic and Electrical Engineering, University of Bath, Bath, UK
[25]National Key Laboratory of Electromagnetic Environment, China Research Institute of Radiowave Propagation, Qingdao, China

**Correspondence:** gunter.stober@unibe.ch

**Abstract.** The Hunga Tonga-Hunga Ha'apai volcano erupted on 15th January 2022, launching Lamb waves and gravity waves into the atmosphere. In this study, we present results using 13 globally distributed meteor radars and identify the volcanic-caused gravity waves in the mesospheric/lower thermospheric winds. Leveraging the High-Altitude Mechanistic General Circulation Model (HIAMCM), we compare the global propagation of these gravity waves. We observed an eastward propagating gravity wave packet with an observed phase speed of 240±5.7 m/s and a westward propagating gravity wave with an observed phase speed of 166.5 ±6.4 m/s. We identified these waves in the HIAMCM and obtained very good agreement of the observed phase speeds of 239.5±4.3 m/s and 162.2±6.1 m/s for the eastward and the westward waves, respectively. Considering that HIAMCM perturbations in the mesosphere/lower thermosphere were the result of the secondary waves generated by the dissipation of the primary gravity waves from the volcanic eruption affirms the importance of higher-order wave generation. Furthermore, based on meteor radar observations of the gravity wave propagation around the globe, we estimate the eruption time to be within 6 minutes of the nominal value of 15th January 2022 04:15 UTC and localized the volcanic eruption to be within 78 km relative to the World Geodetic System 84 coordinates of the volcano confirming our estimates to be realistic.

## 1 Introduction

The Hunga Tonga-Hunga Ha'apai (HTHH) volcano erupted on 15 January 2022, with the strongest eruption occurring at 04:15 UTC. This eruption injected a gigantic amount of water vapor into the stratosphere and mesosphere (Millán et al., 2022), generated a ash plume reaching up to 57 km in altitude (Carr et al., 2022), and launched Lamb waves and gravity waves into the atmosphere (Wright et al., 2022; Liu et al., 2023; Vadas et al., 2023a; Watanabe et al., 2022). Because of the very large amplitudes involved, this eruption provided a unique opportunity to study the gravity wave propagation around the globe throughout all atmospheric layers from the troposphere to the thermosphere and ionosphere. Many previous studies focused on the Lamb waves generated by the sudden and vigorous explosion of the volcano in the Pacific (Wright et al., 2022; Matoza et al., 2022; Liu et al., 2023). Total Electron Content (TEC) observations around the Pacific region indicated strong ionospheric disturbances associated with the Hunga-Tonga eruption (Themens et al., 2022; Heki, 2022; Yamada et al., 2022; Zhang et al., 2022). Recent modeling and observations indicate that these disturbances seemed to be the result of secondary gravity wave generation rather than a direct propagation of the Lamb wave for which the upper mesosphere is an evanescent region due to the lower speed of sound (Vadas et al., 2023b; Stober et al., 2023). The more complicated propagation conditions in the mesosphere with the extremely cold mesopause and highly variable winds indicate why observations of the Hunga-Tunga eruption are under-represented in the literature for this atmospheric layer. Wright et al. (2022) reported a signature of the Hunga-Tonga-caused waves over Hawaii seen in airglow. Meteor radar observations involving the Nordic Meteor Radar Custer and the Chilean

Observation Network De Meteor Radars (CONDOR) indicated that the Lamb waves from the HTHH were attenuated in the mesosphere/lower thermosphere (MLT) and that the strongest amplitudes and signatures of the volcanic eruption were found as eastward and westward propagating gravity waves with intrinsic phase speed of about 210 m/s and horizontal wavelengths of 1600-2000 km (Stober et al., 2023).

Simulations of the TEC disturbances in the ionosphere and thermosphere reveal how MLT winds affect the upper atmosphere through $E \times B$-coupling (Miyoshi and Shinagawa, 2023; Shinbori et al., 2022). Such multistep vertical coupling, which results from the dissipation of primary GWs from HTHH in the thermosphere is important for explaining why the volcanic-caused waves have much higher phase speeds in TEC observations (Themens et al., 2022; Vadas et al., 2023b) and in MIGHTI (Michelson Interferometer for Global High-Resolution Thermospheric Imaging) neutral wind measurements in the thermosphere (Vadas et al., 2023a) as compared to the waves reported in the lower and middle atmosphere (Wright et al., 2022; Matoza et al., 2022; Stober et al., 2023). The HTHH event also provided a benchmark for the modeling of the volcano-triggered gravity waves and their global propagation in a variable background wind field. Simulations with the Ground-to-Topside Model of Atmosphere and Ionosphere for Aeronomy (GAIA) and the Whole Atmosphere Community Climate Model With Thermosphere and Ionosphere Extension (WACCM-X) modeled the GW and pseudo-Lamb waves and their propagation around the globe (Miyoshi and Shinagawa, 2023; Liu et al., 2023). In this study, we leverage the Model for gravity wavE SOurce, Ray trAcing and reConstruction (MESORAC)/High Altitude Mechanistic Circulation Model (HIAMCM) and demonstrate that concentric gravity wave structures generated by the HTHH eruption reached the thermosphere causing perturbations in the neutral winds. Other studies based on the above-mentioned models showed that the volcanic signature of the eruption also inhibited disturbances in TEC (Miyoshi and Shinagawa, 2023; Vadas et al., 2023b, a) which are well supported by co-incident TEC observations.

In the present paper, we investigate the mesospheric propagation of gravity waves resulting from the HTHH eruption using globally distributed meteor radar (MR) observations covering the latitudes from 79°N on Svalbard to 78°S at McMurdo, Antarctica. We identified these gravity waves in wind measurements using 13 different meteor radars at various distances and azimuths from the eruption site, which allows us to infer the observed wave phase speeds for eastward and westward propagation with high accuracy. Furthermore, we demonstrate some asymmetries in the concentric gravity wave propagation, which led to characteristic amplitude variations in both the horizontal wind components as well as the vertical at certain stations. Furthermore, we perform a similar analysis using HIAMCM wind data to compare the observational results with the model wind fields (Vadas et al., 2023a).

## 2  Meteor radar observations

Meteor radars have been used for decades to measure the winds and meteor flux in the MLT (Hocking et al., 1997, 2001; Holdsworth et al., 2004; Meek et al., 2013). The HTHH eruption presented an opportunity to demonstrate the capabilities of this technique to detect and trace GWs from a known source around the globe using a common and standardized analysis.

The applied retrievals provide information about both horizontal wind components and also infer the vertical wind component using a Tikhonov regularization to account for the low measurement response (Rodgers, 2000; Stober et al., 2022). On the

other hand, the biggest challenge in processing the data from monostatic single station data is to achieve a temporal resolution of 10 min while still keeping a good altitude coverage of at least 10 km over the meteor layer. Considering the long vertical wavelength of the HTHH GWs, we kept the frequently used vertical resolution of 2 km but with a 5-kilometer vertical averaging window centered around the respective altitude, including a Gaussian weighting (Hindley et al., 2022), leading to an effective vertical resolution of approximately 4 km (Stober et al., 2023). On the other hand, a temporal averaging of more

than 10 minutes substantially weakens the volcanic gravity wave signature and, thus, reduces the probability of identifying the HTHH GWs in the wind measurements.

We identified the HTHH GWs using measurements from 13 meteor radars around the globe. The meteor radars are located at McMurdo (Marino et al., 2022), Davis, Rothera (ROT) (Dempsey et al., 2021), in Argentina the Southern Argentina Agile Meteor Radar (SAAMER) Tierra del Fuego (TDF) (Fritts et al., 2010), CONDOR represented by the Andes Lidar Observatory

(ALO) (Stober et al., 2021b), Cariri (CAR) (Andrioli et al., 2013), Canadian Meteor Orbit Radar (CMOR) (in this study also - CMO) (Webster et al., 2004; Brown et al., 2010), Pokerflat (PKF), Mengcheng (MEN) (Yang et al., 2023), Kunming (KUN) (Zeng et al., 2022), Svalbard (SVA) (Hall and Tsutsumi, 2020), the Nordic Meteor Radar Cluster (NORDIC) represented by the Tromsø (TRO) (Hall and Tsutsumi, 2013; Stober et al., 2021b) and Collm (COL) (Jacobi et al., 2007). The Nordic Meteor Radar Cluster data was reduced to be equivalent to a monostatic meteor radar located at the reference coordinate, which is

close to Tromsø. All monostatic meteor radars were analyzed by applying a retrieval algorithm including the World Geodetic Reference System (WGS84, (National Imagery and Mapping Agency, 2000)), to compute the East-North-Up coordinates at the geographic location of each detected meteor. Non-linear error propagation was considered to incorporate the spatial and temporal wind shears within each time-altitude bin (Gudadze et al., 2019; Stober et al., 2021a). More details of the implementation can be found in Stober et al. (2021b, 2022). Figure 1 shows a map visualizing all meteor radar locations. The blue

lines indicate the shortest great circle (GC) distance between each meteor radar and HTHH. The dashed red line indicates the full GC path connecting the volcano and ALO, CAR, MEN, and KUN. The yellow line indicates the long GC path along the eastward direction connecting HTHH to the European radars. The lines indicate idealized propagation paths of the secondary GWs caused by the eruption and are used to measure the distances. The distances shown in the map are computed at 90 km altitude to account for the mesospheric propagation.

The meteor radars used in this study provide good coverage around the Pacific to investigate potential differences or asymmetries of the volcano-triggered GWs. Furthermore, four stations, namely MEN, KUN, ALO, and CAR are located very close to the same GC (dashed red line) and, thus, it is straightforward to infer differences in the eastward and westward propagation of the GWs. A summary of all stations with their geographic coordinates and their eastward and westward GC distances from the HTHH eruption site are summarized in Table 1.

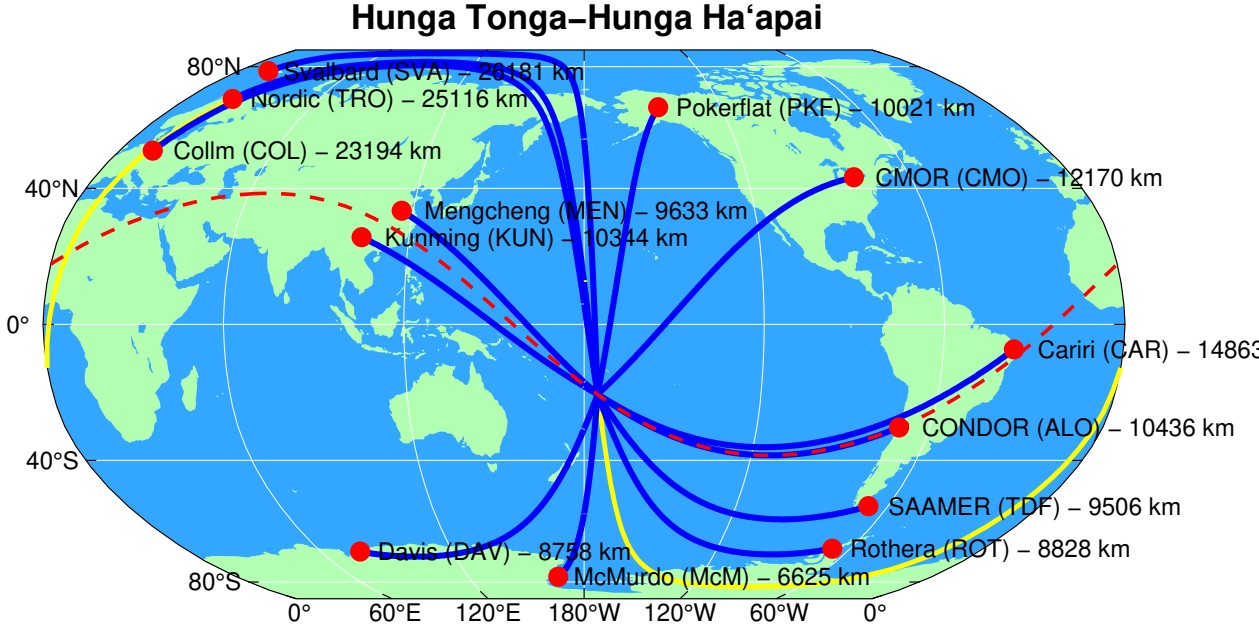

**Figure 1.** Robinson projection of the globe centered on the Pacific. The solid blue lines show the shortest GC distance between the volcanic eruption and the center of each station. The solid yellow line reflects the long GC path for the European radars. The dashed red line indicates the full GC path connecting the volcano and ALO, CAR, MEN, and KUN.

## 3   HIAMCM global fields

The HIAMCM is a gravity wave-resolving General Circulation Model (GCM) from the surface to about 400 km altitude. HIAMCM has a triangular spectral truncation at a horizontal wavenumber 240, which results in an apprximate grid spacing of 55 km and a resolved horizontal wavelength of $\lambda_h =$165 km. The vertical spacing changes with altitude. From the boundary layer up to $z = 100$ km corresponding to $p = 3 \times 10^{-4}$ hPa) the vertical level spacing is 600 m. In the thermosphere, the vertical grid resolution increases to about 5 km at 300 km altitude. In total, there are 261 full-model layers with the highest layer at $p = 6 \times 10^{-9}$ hPa at approximately $z = 400$ km. A detailed description of the HIAMCM is found in Becker and Vadas (2020). Recently, it was demonstrated that the HIAMCM can be nudged to meteorological reanalysis such as The Modern-Era Retrospective Analysis for Research and Applications (MERRA-2). The nudging is implemented in the spectral domain, which allows to specify realistic large-scale meteorological fields through which the resolved GWs propagate (Becker et al., 2022).

Leveraging these new capabilities, the primary and secondary GWs resulting from the HTHH eruption were simulated with the Model for gravity wavE SOurce, Ray trAcing and reConstruction (MESORAC) and the HIAMCM, respectively, and were analyzed to study the impact of the volcanic eruption on the thermospheric-ionospheric system (Vadas et al., 2023b, a). The MESORAC calculates the primary GWs created from localized (in space and time) vertical updrafts of air using the Fourier-Laplace analytical fully-compressible solutions (Vadas, 2013). These up-drafts, which are mechanical displacements

| | latitude | longitude | GC (short) / km | GC (long) / km | references |
|---|---|---|---|---|---|
| McM (McMurdo) | 77.85°S | 166.72°E | 6626 (W) | 33404 (E) | Marino et al. (2022) |
| DAV (Davis) | 68.58°S | 77.97°E | 8759 (W) | 31271 (E) | |
| ROT (Rothera) | 67.57°S | 68.12°W | 8828 (E) | 31202 (W) | Dempsey et al. (2021) |
| TDF (Tierra del Fuego) | 53.79°S | 67.75°W | 9505 (E) | 30525 (W) | Fritts et al. (2010) |
| MEN (Mengcheng) | 33.4°N | 116.0°E | 9634 (W) | 30398 (E) | Yang et al. (2023) |
| KUN (Kunming) | 25.6°N | 103.0°E | 10345 (W) | 29685 (E) | Zeng et al. (2022) |
| ALO (CONDOR) | 30.3°S | 70.7°W | 10436 (E) | 29594 (W) | Stober et al. (2021b) |
| CAR (Cariri) | 7.38°S | 36.53°W | 14862 (E) | 25168 (W) | Andrioli et al. (2013) |
| PKF (Pokerflat) | 65.13°N | 147.5°W | 10020 (E) | 30010 (W) | |
| CMO (CMOR) | 43.26°N | 80.77°W | 12169 (E) | 27861 (W) | Webster et al. (2004) |
| | | | | | Brown et al. (2010) |
| SVA (Svalbard) | 78.17°N | 15.99°E | 13848 (W) | 26182 (E) | Hall and Tsutsumi (2020) |
| TRO (Nordic) | 67.9°N | 21.1°E | 14913 (W) | 25117 (E) | Hall and Tsutsumi (2013) |
| | | | | | Stober et al. (2021b) |
| COL (Collm) | 51.31°N | 13.0°E | 16836 (W) | 23194 (E) | Jacobi et al. (2007) |

**Table 1.** Summary of the geographic locations for each meteor radar and GC distances relative to HTHH (20.54°S, 175.38°W). The term short/long refers to the shortest/longest distance along the GC. The brackets after the GC distance indicate whether the westward or eastward HTHH GWs reached the station along this path.

of stratospheric/mesospheric air, are identified from NOAA's Geostationary Operational Environmental Satellite (GOES) data. The atmosphere responds by radiating concentric rings of GWs (Vadas and Liu, 2009; Vadas et al., 2012). MESORAC ray traces these GWs forward in time, including their phases, and reconstructs the primary GW field using the GW phases and the GW dissipative dispersion and polarization relations (Vadas and Fritts, 2005, 2009). The background atmosphere is taken from the HIAMCM simulation for 15 January 2022 without the Tonga eruption (base case) using scales with $\lambda_h > 2000$ km.

Wave dissipation is due to molecular diffusion and turbulent diffusion from saturation. The body forces and heatings created by the dissipation of primary GWs are calculated as functions of space and time as in Vadas (2013). These ambient-flow effects are then added to the momentum and thermodynamic equations of the HIAMCM to simulate the secondary GWs from the Tonga eruption (Vadas et al., 2023a). The turbulent diffusion coefficient $D_0$ is tuned to 2,000 $m^2$/s because it results in GW amplitudes that are closest to the ICON-MIGHTI large-scale amplitudes without destabilizing the HIAMCM. Larger values of

$D_0$ result in smaller GW amplitudes, which is inconsistent with the ICON data.

In this study, we focus on the mesospheric data from the HIAMCM between 80-100 km and with a temporal resolution of 5 minutes using the model runs from Vadas et al. (2023a). The simulated perturbations are defined as the differences between the results from the "Tonga run" (with inputs from MESORAC) and the "base run" (without inputs from MESORAC). The HTHH GWs are extracted by subtracting a reference run from the disturbance simulation. Furthermore, we apply an observational filter

for each meteor radar to account for the spatial and temporal sampling of these radars similar to previous studies (Pokhotelov

et al., 2018; Stober et al., 2020, 2021c).

Figure 2 shows maps of how the HTHH secondary GWs propagated around the Earth at 4 different times on the 15th of

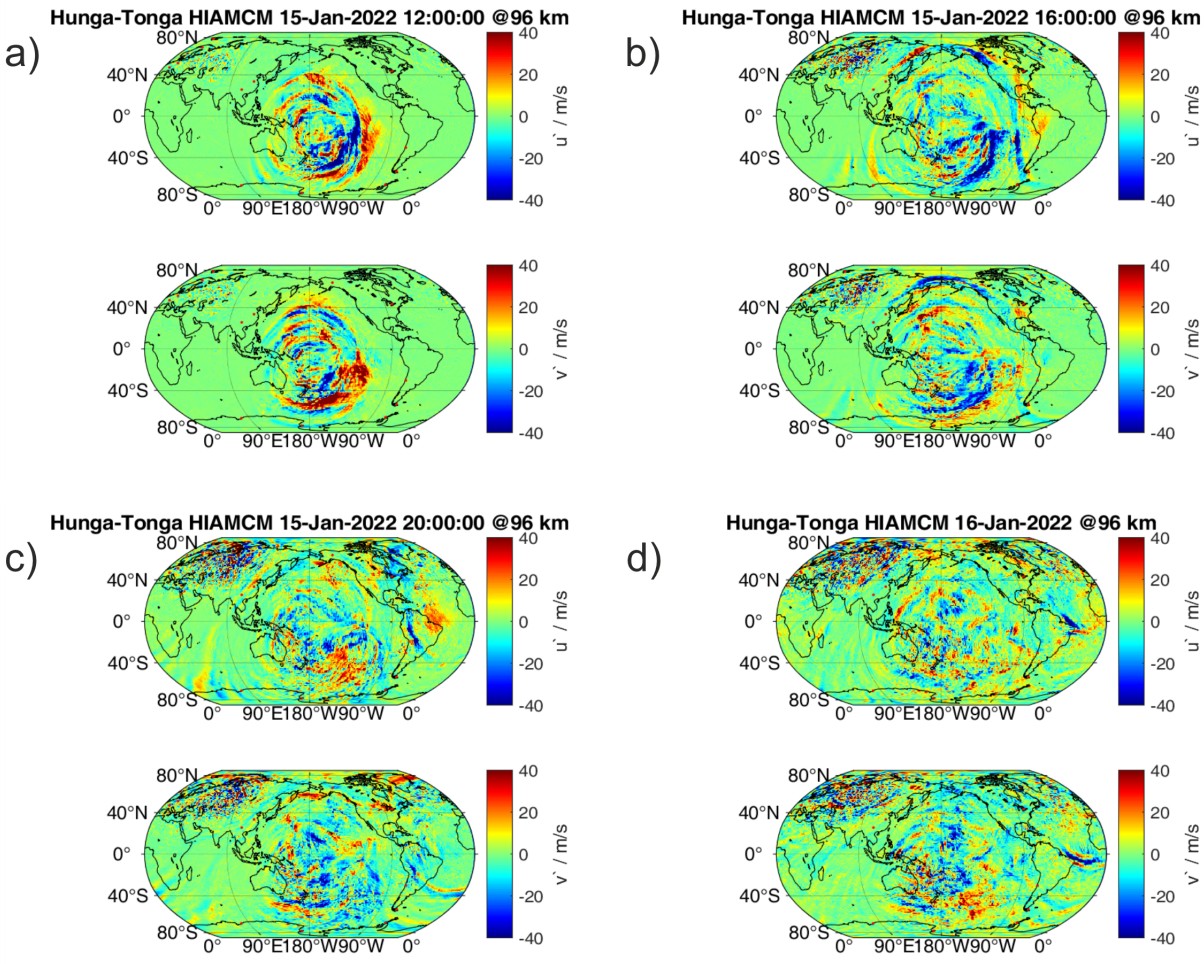

**Figure 2.** HIAMCM zonal wind perturbations $u'$ (upper panel) and meridional wind perturbations $v'$ (lower panel) from 15th January 2022 at 12, 16, 20, and 24 UT (a-d).

January 12 UT, 16 UT, 20 UT, and 24 UT. Note that each panel in Fig. 2 consists of an upper and a lower part that shows the zonal and the meridional winds, respectively. The HIAMCM GWs propagate radially away from the HTHH eruption site. However, the model also reflects a substantial radial asymmetry in the GW amplitudes indicating that the eruption resulted in different wavefronts for different azimuthal directions. This asymmetry is also reflected in the Keograms for the zonal and meridional wind at the longitude and latitude of HTHH, which are shown in Figure 3 and Figure 4. The Keograms visualize the propagation of the HTHH GW perturbations with latitude and longitude. The black lines show the theoretical lines assuming a phase velocity of 240 and 165 m/s.

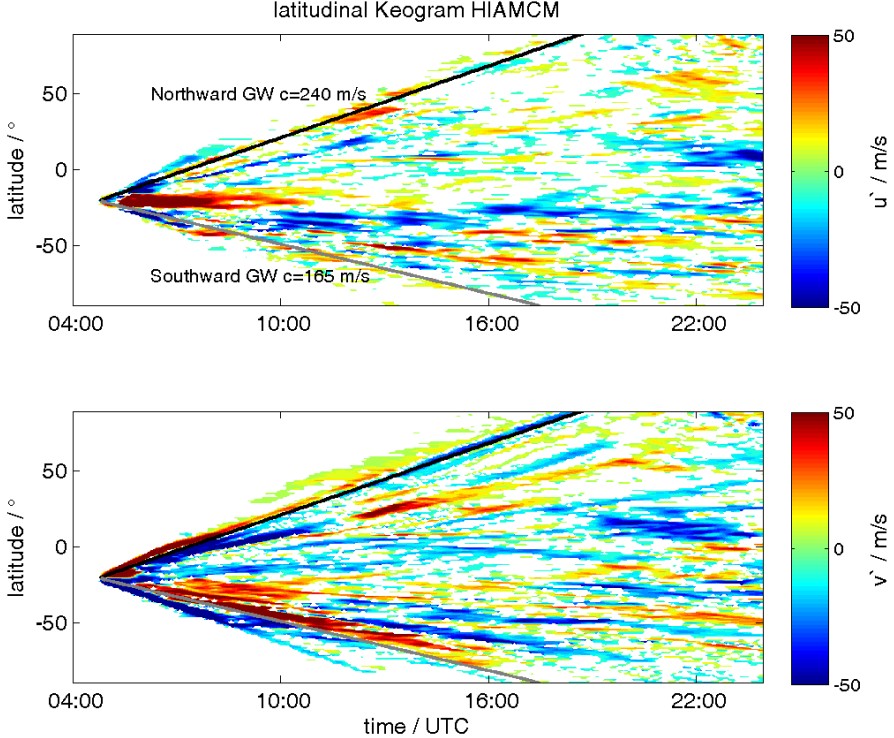

**Figure 3.** Latitudinal Keograms of the zonal and meridional wind perturbations from HIAMCM for the 15th January 2022. The reference latitude and longitude are the geographic coordinates of HTHH. The black lines show the theoretical lines assuming a phase velocity of an eastward propagating HTHH GW with 240 m/s and a westward HTHH GW with 165 m/s.

The largest amplitudes are found for the southeastward propagation (towards the Andes and the Antarctic Peninsula), whereas much smaller GW amplitudes were simulated over Australia, China, and the North American sector. This asymmetry (of the secondary GWs) is caused by the north and southward orientation of the local body forces (i.e., horizontal accelerations) in the thermosphere that are generated by the localized deposition of momentum that occurred at z 120-180 km where the primary GWs from Tonga broke and dissipated (Vadas et al., 2023a). These primary GWs were mainly propagating

meridionally when they dissipated and created the body forces because the background wind was strongly meridional in the thermosphere at the time. It is well-known that a local horizontal body force generates an asymmetric GW distribution, with the maximum amplitudes being in and against the body force direction, and with zero amplitudes perpendicular to the body force direction (Vadas et al., 2003; Vadas and Becker, 2018; Themens et al., 2022).

    In addition, some of the additional azimuthal GW asymmetries may be due to the complex eruption sequence (Vadas et al.,

2023a). Results from the HIAMCM also show that the HTHH GWs traveling westward along the GC towards Europe are disturbed by the polar vortex in the northern hemisphere and, thus, almost no coherent wavefront arrives over Scandinavia,

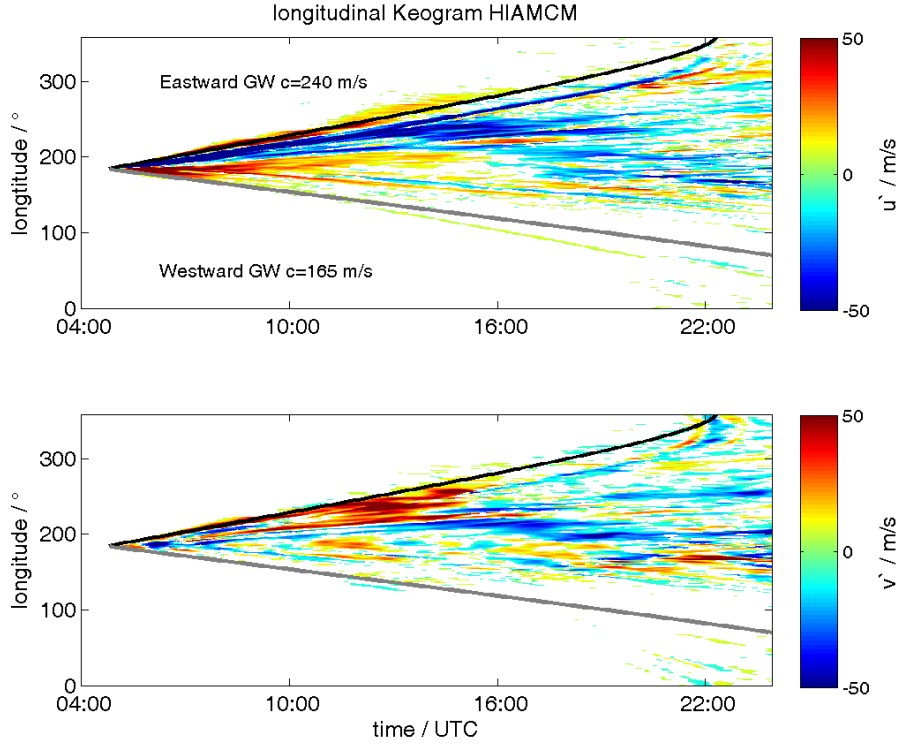

**Figure 4.** Longitudinal Keograms of the zonal and meridional wind perturbations from HIAMCM for the 15th January 2022. Otherwise the same as Figure 3.

although this would have been the shortest distance to HTHH. On the other hand, the HIAMCM results suggest that the most coherent wavefront from HTHH propagated towards South America and even gained strength over the Atlantic Ocean before reaching central Europe. This is consistent with the observations presented in Stober et al. (2023) and is related to the geometric refocusing of rays propagating on a sphere and approaching the mid-point to the antipode in North Africa (Matoza et al., 2022; Amores et al., 2022).

## 4   Comparison of HIAMCM results and meteor radar observations

Tracking the HTHH secondary GWs around the globe from meteor radar wind observations requires identifying the whole GW packet rather than one distinct feature that can be found in all observations. The main eruption lasted 3 hours and consisted of many vigorous detonations resulting in updrafts (Vadas et al., 2023a) triggering the HTHH primary GW field. The secondary GWs were generated by the local body forces created by the dissipation of the primary GWs in the mesosphere and thermosphere. Considering the HIAMCM simulation of the HTHH GW packet illustrates that there is a huge variety of

possible amplitudes and shapes of the GW packet depending on the different GC paths along which the waves are propagating. Furthermore, derived phase velocities will depend on the ability to detect the leading edge of the HTHH GW packet, rather than on the maximum amplitude, which often corresponds to a later time of the eruption sequence, to identify the correct arrival time in the observation volume. Thus, the time of arrival estimates are prone to large uncertainties that have to be considered in the further analysis of HTHH GWs. In the following, we briefly outline the analysis procedure.

We examined the 10 min resolution meteor radar winds and subtracted a 4-hour running window to remove traveling planetary waves (mostly the Quasi-2-Day-Wave (Q2DW) in the southern hemisphere), atmospheric tides, and GWs with observed periods longer than 4 hours. The vertically integrated anomalies were then inspected to identify the HTHH GW packet by searching for peaks in the zonal, meridional, and total horizontal wind speed. Our observations can be grouped into three different classes depending on the shape of the GW packet, the amplitude, or a superposition with other waves. The timing was then estimated for all stations that showed a sinusoidal wave by the onset of the GW packet. Typical examples of these stations are ALO, PKF, and TDF. For the other stations, we searched for the first GW amplitude peak in an interval around the estimated arrival time e.g., CMOR, McM, and DAV.

Figure 5 compares the theoretical arrival times derived from the best fit over all stations along a GC path of the wave packet leading edge for a subset of meteor radars (left column) and HIAMCM winds from Tonga run minus base run ( right column) from the Antarctic Peninsula to Alaska. The time of the first eruption is shown as a yellow vertical line. The cyan vertical lines indicate the arrival times of the eastward propagating HTHH GW packet estimated from the best-fit phase speed using all stations that detected this wave, whereas the black vertical lines show the best fit of the arrival time of the westward wave. The dashed vertical red lines show the one-sigma range of the leading edge arrival time for each station. All vertical lines enclose a time window of 3 hours corresponding to the time of the major eruption sequence. The y-axis for the HIMACM winds is reduced by a factor of 2.8 to mitigate the smaller wind perturbations in the model. The observations and the HIAMCM show a remarkable agreement for several stations in both the timing and wave pattern at ALO, PKF, or CAR. Only CMOR exhibits a visible difference in the observed arrival time compared to HIAMCM. Although CMOR is one of the most powerful systems, concerning the number of meteors detections and wind altitude coverage used in this study, the HTHH GW is barely identifiable in the zonal wind component; the observations indicate a much less clear wavefront and an earlier arrival time compared to the model and concerning the other stations. We note that the diurnal variation of the meteor count rate has an impact on the altitude coverage as well as the statistical uncertainties of the retrieved winds. KUN, MEN, and CAR are most significantly affected by the diurnal count rate.

 The comparison of the wind anomalies of the Antarctic stations of McM and DAV and the two radars located in China (MEN and KUN) are presented in Figure 6. Although the two meteor radars on the Antarctic continent are located at the closest distance to HTHH, the GW signature is barely visible in the data. The HTHH GW is mainly identified by a coherent sinusoidal signature (between the vertical lines), which was also found in other stations. However, the amplitude of this wave signature does not exceed the background atmospheric signal for these two stations. Furthermore, these two stations also reflect the largest discrepancy in the arrival time between the observations and HIAMCM (see section 7). Interestingly, KUN and MEN meteor radars observed the HTHH GW with a much smaller amplitude than their counterparts eastward of the volcano in

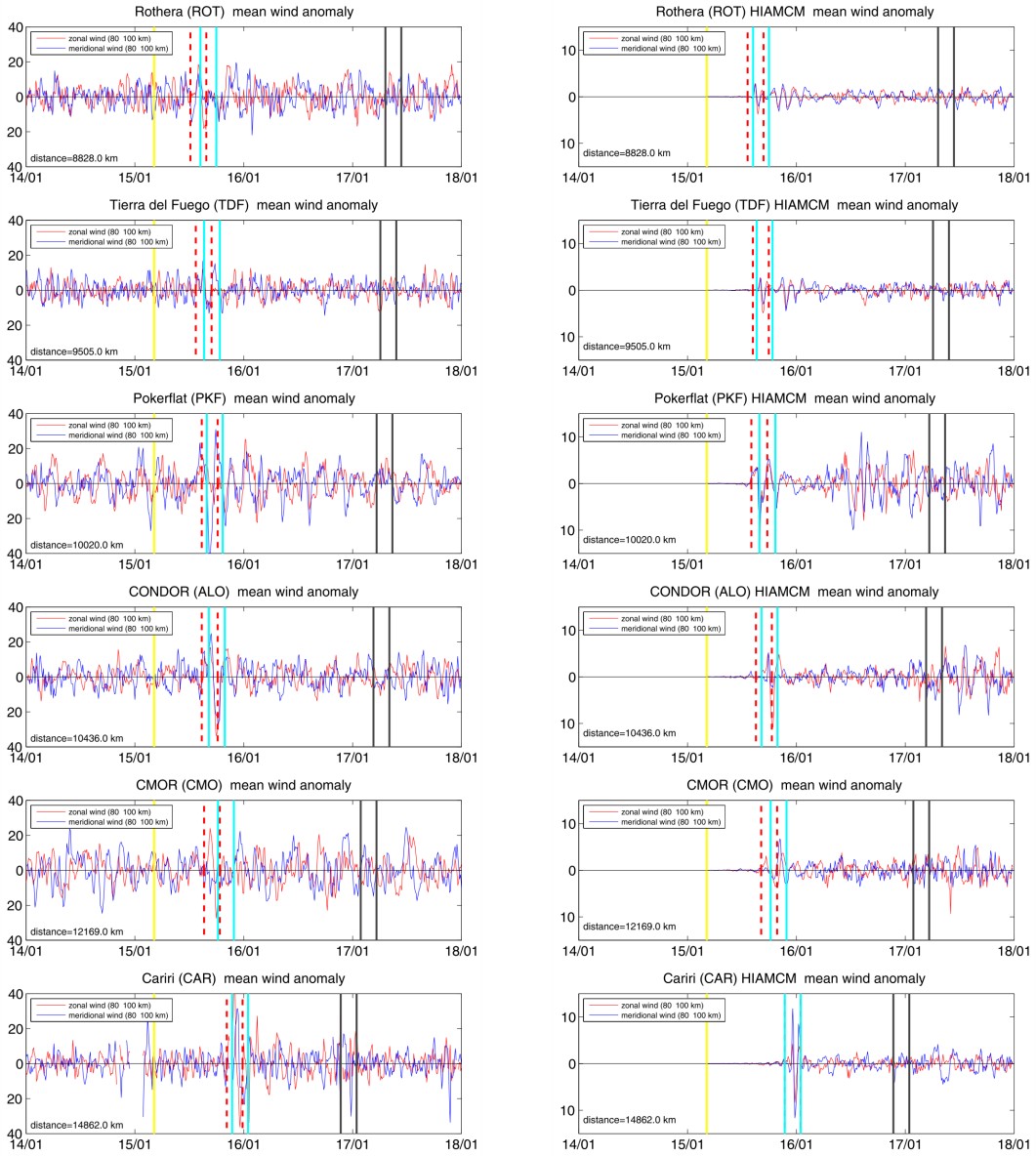

**Figure 5.** Wind anomalies for the meteor radars located eastward of HTHH and corresponding anomalies from HIAMCM. The date ticks are given at 00:00 UTC of the date. The time of the volcanic eruption is indicated by a vertical yellow line. The cyan vertical lines span the predicted arrival of the eastward propagating GW. The black vertical lines indicate the timing for the westward propagating HTHH GW packet. The dashed vertical lines indicate the one-sigma interval of the time picks for each station. Note that the HIAMCM shows perturbations only to the right of the yellow line because we plotted the differences between the Tonga run and the base run (see Sec. 3). A higher resolution of the HTHH GW wave packet is found in Appendix A1.

South America (ALO, and CAR), even though all these stations are roughly on the same GC. Our results therefore support the azimuthal asymmetry of the GW zonal and meridional wind perturbations in the HIAMCM. The increased variability in the wind anomalies before the HTHH in the meteor radar data is mainly due to the diurnal variation in the meteor count rate, which reached a minimum during these times. Another important aspect is that the zonal and meridional winds were out of phase for the MEN and KUN stations, indicating a northwestward propagation, whereas CAR and ALO exhibit an in-phase relationship between the two wind components, which is expected for eastward propagating. Also, the arrival times between the HIAMCM and the Chinese meteor radars are in much better agreement compared to McM and DAV.

Wind anomalies above the European sector are shown in Figure 7. We identified the eastward propagating GWs (long GC

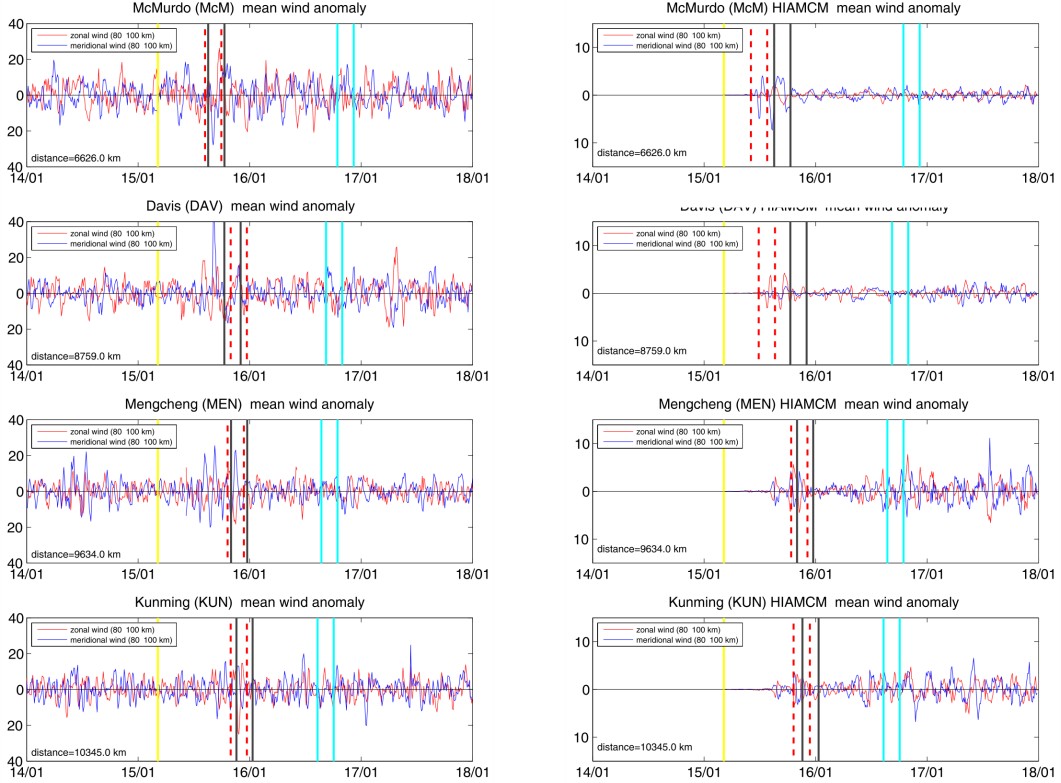

**Figure 6.** The same as Figure 5, but for the stations westward of HTHH outside the European continent. A higher resolution Figure of the HTHH GW wave packet is found in Appendix A2.

path) by searching for identical patterns in the wind anomalies for all three stations. The cyan vertical lines mark a coherent wave structure that exhibits an in-phase behavior of the zonal and meridional winds (as expected since the GWs are propagating northeastward), similar to what was found for all the South American stations. HIAMCM winds also show some remnants of the eastward HTHH GW over the European sector. In particular, the phase behavior between HIAMCM and observations exhibits a remarkable agreement at SVA, TRO, and COL. Overall, the data is much noisier in the northern hemisphere winter

due to GWs from orographic forcing and the polar vortex. The increased variability, compared to pre-HTHH-eruption time, in the observations may be due to the generation of GWs created by seeding of GWs by the Lamb wave that arrived a few hours before the HTHH GWs (Stober et al., 2023). In addition, we note that Lamb waves likely create a continuum of GWs. For the HTHH Lamb waves the seeding process seems to occur at or above z≃110 km (Vadas et al., 2023b).

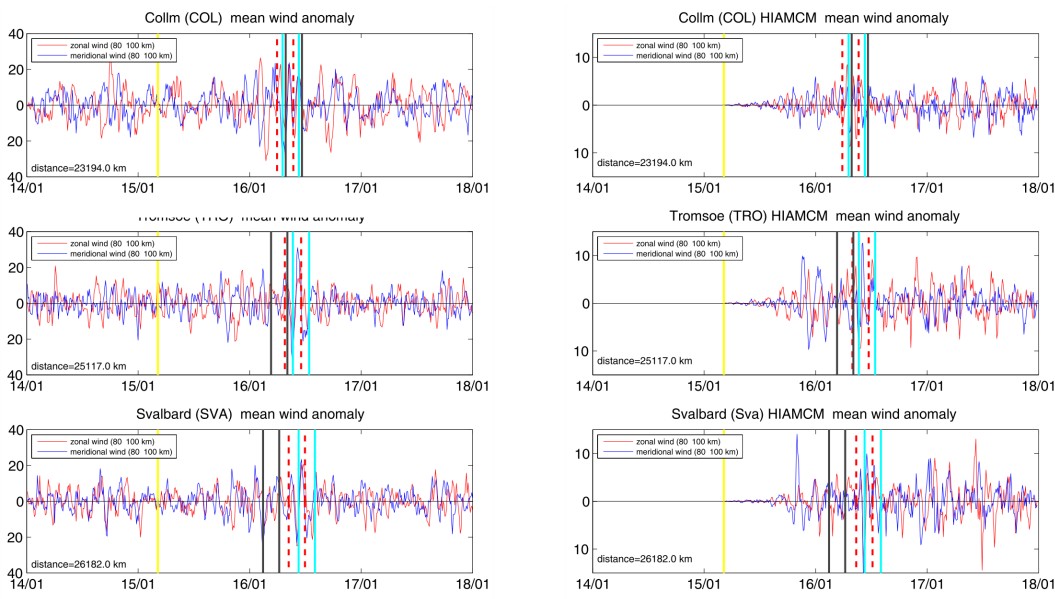

**Figure 7.** The same as Figure 5, but for the European meteor radar sites. A higher resolution Figure of the HTHH GW wave packet is found in Appendix A3.

## 5  Gravity wave observed phase speeds

Gravity wave phase speeds are derived from all meteor radar stations using the times of arrival for each station. The phase speeds of the GW packet with the largest amplitude are determined by fitting a distance vs. time and time vs. distance to account for errors in the time estimates as well as for the sizes of the observation volumes. We assumed a 3-hour uncertainty for the time corresponding to the most active part of the eruption sequence. The spatial uncertainty is given by the diameter of the observation volume and we used 350 km for all stations.

Figure 8 shows four panels summarizing the analysis. The left column presents the meteor radar observations and the right column presents the HIAMCM data. The observational data can be grouped in an eastward wave (red line) with an observed phase speed of 240±5.7 m/s and a westward traveling GW (blue line) exhibiting an observed phase speed of 166±6.4 m/s. The fits were obtained by separating the stations according to their relative position (eastward or westward) of HTHH. The eastward stations of ROT, TDF, ALO, PKF, CMOR, CAR, COL, TRO, and SVA provide enough data to determine the eruption time to be within 6-12 min and eruption location to be within 80 km of the nominal HTHH coordinates in the WGS84 reference.

The precision of the fitting coefficients significantly depends on the European stations. Due to the much larger distance and time between the eruptions, the errors of the individual measurements are less critical. The westward wave (blue) was analyzed using a regularized fit. The fit is regularized by a fixed eruption time $t_0 = 0$ hours and by a fixed location using the nominal HTHH coordinate. This strategy is necessary to account for the much smaller number of observations. The relative distance between DAV, KUN, and MEN and the volcano is almost identical leaving essentially only McM as a second point. Thus, we added the westward HTHH wave signature that was found in the ALO and already presented in Stober et al. (2023) to the fit. The same analysis is applied to HIAMCM and the fitted observed phase velocities for the eastward and westward HTHH GW are in very good agreement between the model and the observations. Only the Antarctic stations of McM and DAV show a substantial deviation. The Q2DW may explain this discrepancy and is clearly visible in the southern hemisphere at mesospheric altitudes in the meteor radar data (Stober et al., 2023) (see section 7). However, there is no indication of this wave found in MERRA2 and, thus, due to the nudging of the large-scale dynamics in HIAMCM from re-analysis, it is also missing in the HIAMCM winds. The amplitude of the Q2DW is strongest in the meridional wind component and showed a clear northward wind direction during the time of eruption above the South American continent.

Finally, Figure 9 compares the derived observed phase speed for each station separately relative to the distance along the GC path from the radars (left panel) and the HIAMCM (right panel). The ALO station is added with both detections (eastward and westward GW packet) of the HTHH GW and, thus, appears at two distances representing the eastward and westward GC distance. There is a remarkable agreement for ALO, SVA, TRO, CAR, COL, MEN, and KUN to within ±5 m/s when comparing the HIAMCM to the observations. Furthermore, this comparison confirms that there are two clusters that correspond to the eastward (240 m/s) and westward (160 m/s) HTHH GW packets. It is also obvious that the observed phase speeds remain roughly constant with distance from the volcano. Possible reasons for the exceptional behavior of DAV and McM in the HIAMCM data have already been discussed above.

## 6  Polarization of HTHH GW along the Great Circle - CONDOR and Mencheng

Currently, there are contradicting interpretations of the meteor radar winds documented in the literature. Poblet et al. (2023) attributed the strong MLT wind perturbation found above CONDOR to be related to the L1 Lamb wave mode, although both the inferred horizontal wavelength and wave period seem to differ from the modeled L1 Lamb wave in WACCM-X simulations (Liu et al., 2023) by an order of magnitude.

This discrepancy between the attribution of HTHH wind perturbations to Lamb waves and GWs can be resolved using polarization relations. GWs show a distinct polarization relation depending on propagation direction and geographic location. We evaluated the polarization relation for GW along the same GC at approximately the same distances for ALO and MEN. The GW polarization relation is given by Fritts and Alexander (2003);

$$\tilde{v} = \frac{(i \cdot \hat{\omega} \cdot l + f \cdot k)}{(i \cdot \hat{\omega} \cdot k - f \cdot l)} \cdot \tilde{u} \ . \tag{1}$$

Here $\hat{\omega}$ is the intrinsic angular wave frequency, $k$ and $l$ correspond to the zonal and meridional wavenumbers, respectively, and $f$ denotes the Coriolis parameter for a given latitude. The zonal and meridional monochromatic GW amplitude is given by $\tilde{u}$ and

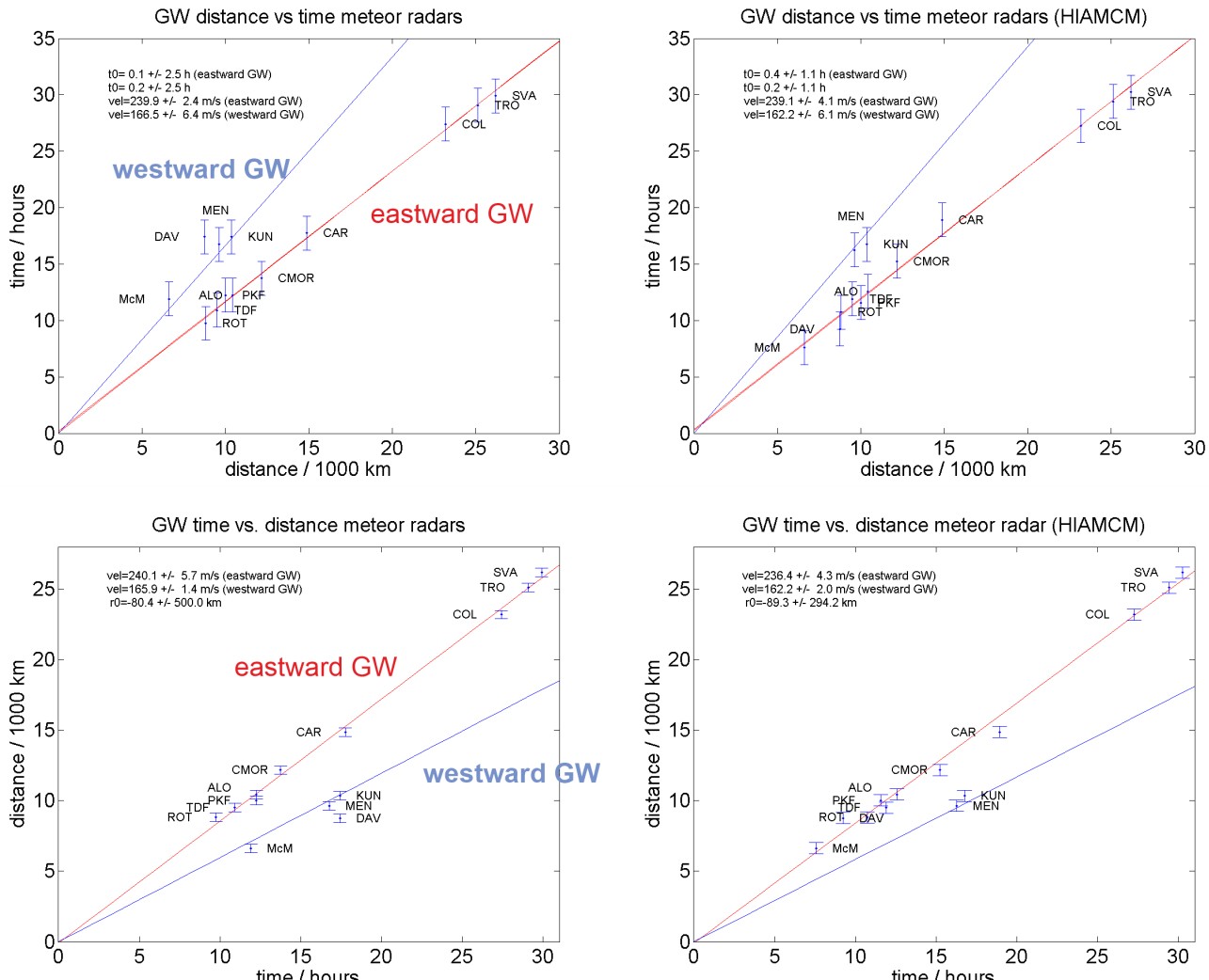

**Figure 8.** Gravity wave phase speed analysis for the meteor radars (left column) and HIAMCM (right column) as distance versa time and time versa distance plots. The red line indicates the best fit for the eastward propagating HTHH GW, the blue line is the best fit for the westward propagating HTHH GW.

$\tilde{v}$. Stober et al. (2023) estimated the horizontal wavelength $\lambda_h = 1600 - 2000$ km and derived an intrinsic wave period of about 2 hours to 2 hours 20 minutes (Due to the long horizontal wavelength the intrinsic wave period is rather close to the observed wave period). Based on these intrinsic wave parameters, we calculated a theoretical phase for the zonal and meridional wind perturbations caused by the HTHH GW at ALO and MEN. At MEN the zonal and meridional wind components are supposed to be out of phase by 160-170°, whereas at ALO both wind components of the leading HTHH GW should be in phase by 5-15°.

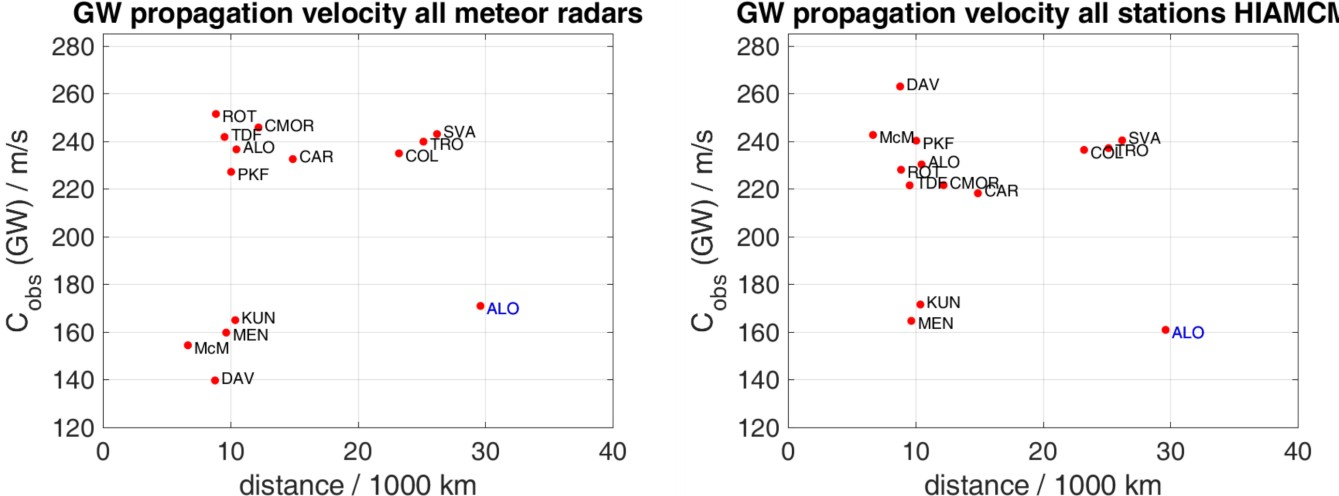

**Figure 9.** Comparison of individual observed phase velocities for the meteor radars and HIAMCM.

Figure 10 shows a comparison of the zonal and meridional wind perturbations from both meteor radars (upper panels) and the corresponding HIAMCM winds (lower panels). The meteor radars winds at MEN exhibit an antiphase relation between both wind components as estimated from the linear theory for this location, whereas at ALO we see a more or less an inphase relation between the meridional and zonal wind perturbations caused by the HTHH GW. This phase behavior is also found in the HIAMCM winds for both stations. A similar agreement is found at KUN. We obtained phase differences between 6-30° for ALO and about 160-180° for MEN, which is very close to the theoretical prediction. Thus, the meteor radar winds exhibit a signature of HTHH GWs that satisfy the polarization relation of GW derived from linear theory (Fritts and Alexander, 2003). The secondary GWs from the HIAMCM reflect this phase behavior. Therefore, we can show from theory and the observed polarization that the observed waves are likely to be GWs and not the L1 Lamb wave.

## 7  The role of the Quasi-two day wave (Q2DW) in the Southern Hemisphere

Q2DWs are common during the hemispheric summer months (Fritts et al., 2019; Iimura et al., 2021). Long-term observations of temperatures from SABER show that the wavenumber 3 dominates in the southern hemisphere (Tang and Gu, 2023). During the HTHH eruption in 2022, the meteor radar winds also exhibited a clear Q2DW periodicity in the background wind. Figure 11 shows the zonal and meridional winds after subtracting atmospheric tides and GW using an adaptive spectral filter (ASF) (Baumgarten and Stober, 2019; Stober et al., 2020). The longitudinal distance between DAV and ROT is $\Delta long = 146.1°$, which roughly corresponds to a wavenumber 3 structure ($\Delta long = 120°$). Although DAV and ROT are almost at the same latitude, only DAV exhibits a Q2DW signature in the zonal wind, whereas ROT reflects the Q2DW in both wind components. The phase difference between the zonal and meridional wind at ROT is $\Delta\phi = 90°$. Based on the phase differences of mean winds from DAV, TDF, ROT, and ALO, we estimate the wavenumber for the Q2DW by producing a simple wave model to

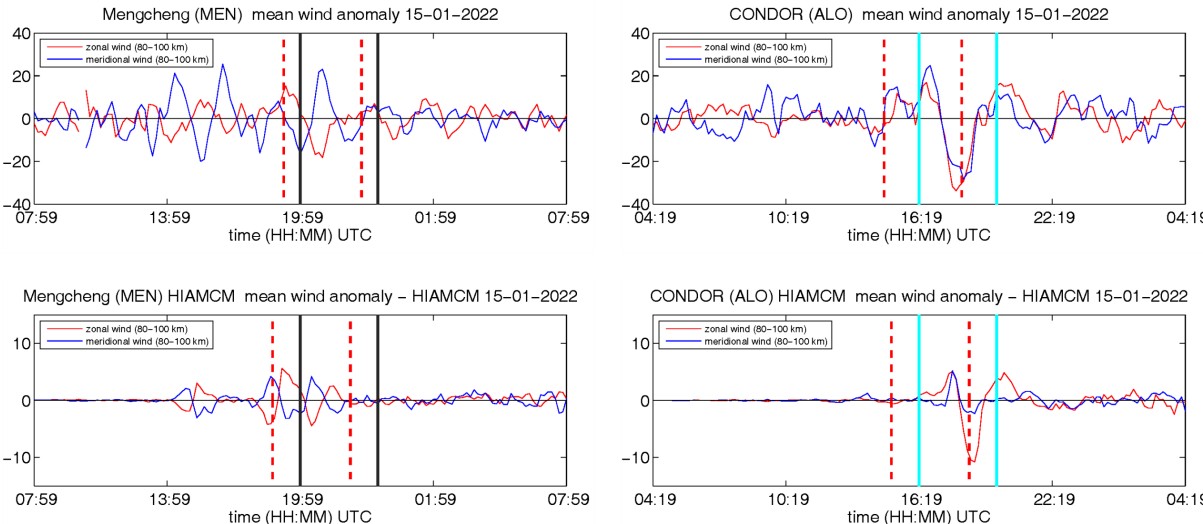

**Figure 10.** Comparison of HTHH GW along the GC connecting HTHH, ALO, and MEN. The upper row shows the meteor radar measurements and the lower row the corresponding HIAMCM perturbation winds.

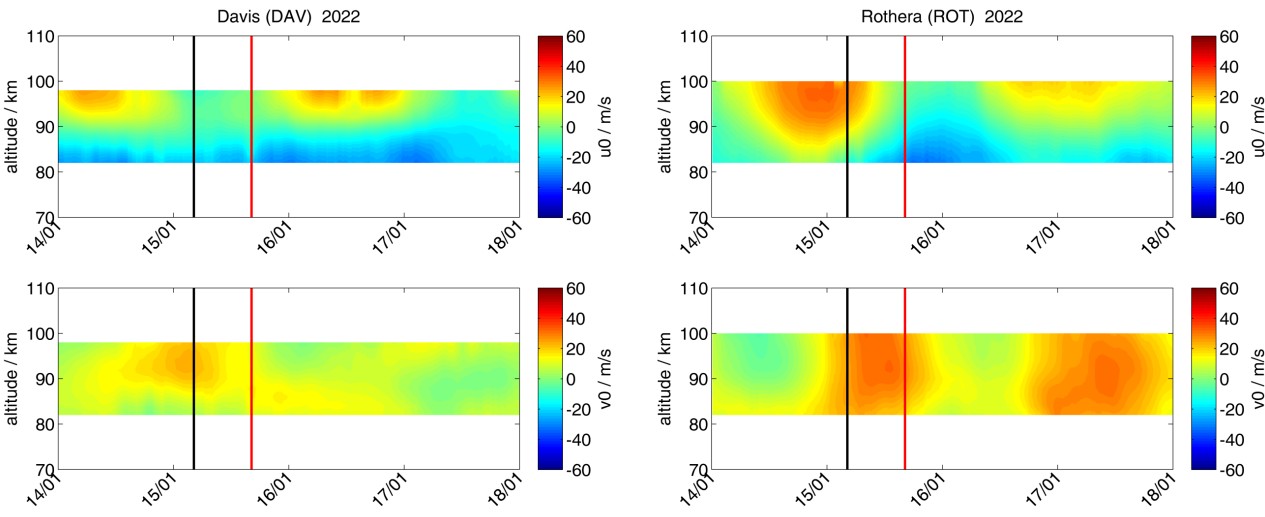

**Figure 11.** Comparison of zonal and meridional mean winds, after subtracting tides and GW with the adaptive spectral filter, above DAV and ROT. The vertical black line indicates the first HTHH eruption at 04:15 UTC on 15th January 2022.

280 account for the latitudinal variability in the Q2DW amplitude in both horizontal wind components. This approach may be more robust given the sparsity of the observations and their latitudinal separation. Similar results are achieved when applying the phase differencing technique directly to DAV and ROT (He et al., 2021). Figure 12 shows an idealized Q2DW with wavenumber 3 projected on the globe for the southern hemisphere covering the latitudes between 70-0°S for the zonal and meridional wind

components, respectively. The left panels show the zonal and meridional wind structure at the time of the first eruption, which is indicated as a black vertical line in Figure 11. The right two panels present the zonal and meridional wind structure upon the arrival of the HTHH GW on the South American continent 12 hours later, which is indicated as a red vertical line in Figure 11. There is also a weak signature of the Q2DW at CAR, which we used to constrain the amplitude at the equator. We examined also other wavenumbers but found no equivalent agreement compared to the wavenumber 3 for the Q2DW during the eruption. The presence of the Q2DW resulted in some interesting effects on the propagation of the HTHH GW. During the eruption,

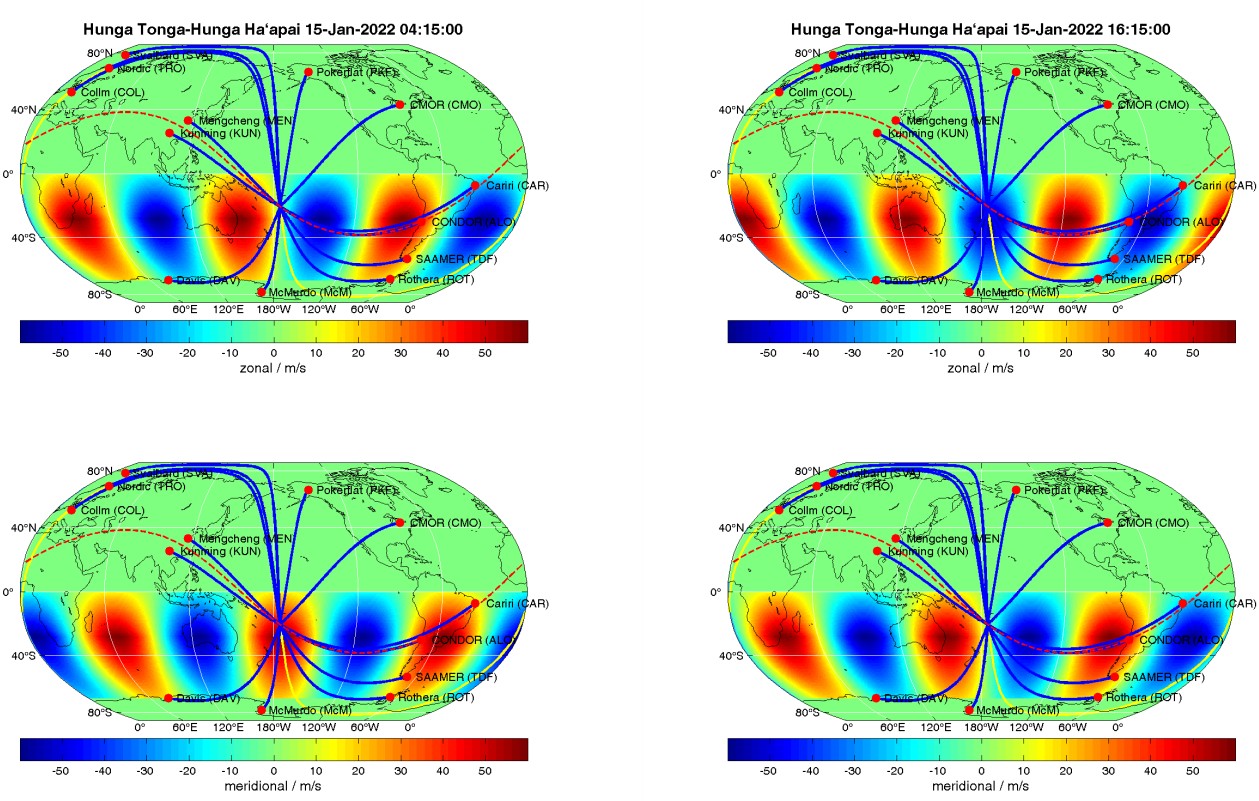

**Figure 12.** Scheme of a westward propagating Q2DW with wavenumber 3. The left two panels show the zonal and meridional wind structure at the time of the eruption. The right column indicates the Q2DW structure when the first HTHH GW arrives at the South American continent approximately 12 hours later. Superimposed on the Q2DW structure, we visualize the GC routes between the HTHH and each meteor radar.

the Q2DW showed a strong northward wind for the location of the volcano. Thus, the HTHH GWs directed towards DAV and McM faced a strong headwind of up to 30-60 m/s along the GC route, which explains the delayed signature of the volcanic GW above the Antarctic meteor radars. Furthermore, the cold polar summer mesopause temperatures at polar latitudes essentially provide a natural barrier and form an evanescent region at the MLT limiting the observed amplitude of the HTHH wave (Vadas

et al., 2023a), although DAV and McM were located at the closest distance to the HTHH. A similar effect on the HTHH GW amplitude is also found at TDF and ROT.

The Q2DW also plays a crucial role in understanding some differences between HIAMCM and the observations for the meteor radars in Antarctica and South America. HIAMCM is nudged to MERRA2 and, hence, the large-scale circulation is prescribed (Becker et al., 2022). Therefore, we examined MERRA2 for the presence of a Q2DW in the horizontal wind components. However, we found no agreement between the zonal and meridional wind that matched the observed Q2DW in the meteor radar data, which explains the differences in the arrival times of the HTHH GWs above DAV and McM between HIAMCM and the meteor radars. Furthermore, the Q2DW also affects the meridional HTHH GW amplitude at ALO.

## 8 Discussion

Although meteor radars have become a widely used sensor to observe mesospheric winds, there have been almost no studies on the HTHH eruption based on such observations. Identifying the HTHH GW in standard monostatic meteor radars turned out to be more challenging than would be expected from the stratospheric data (Wright et al., 2022) and only became feasible due to the results obtained from the high-resolution analysis with the 3DVAR+DIV algorithm (Stober et al., 2023) using MRs with high meteor detection rates such as CONDOR and the Nordic Meteor Radar Cluster. Motivated by this original study, Poblet et al. (2023) aimed to identify the HTHH-caused Lamb wave in hourly winds derived from CONDOR and other South American meteor radar networks. However, such a temporal resolution appears to be insufficient to unambiguously identify the HTHH-Lamb wave considering the wave properties presented in Liu et al. (2023). The HIAMCM simulations used for comparison to the radar observations are found to produce a much more fragmented wavefront for the volcanic-caused GWs, including different GW amplitudes for different azimuths relative to the eruption site (Vadas et al., 2023a) when compared to a high-resolution WACCM-X simulation of the pseudo-Lamb wave modes (Liu et al., 2023).

Previous publications emphasized the Lamb waves generated during the first eruption (Wright et al., 2022; Matoza et al., 2022), which was observable in surface pressure data and stratospheric brightness temperatures observed by satellites. However, high-resolution wind measurements indicated only weak signatures of a possible Lamb wave in the MLT below 90 km altitude (Stober et al., 2023). Upon reaching the GW seeding altitude of $z\simeq110$ km, the small-amplitude Lamb waves could have seeded a continuous spectrum of upward and downward-propagating weak-amplitude GWs similar to the GW spectrum excited at the surface by a localized ocean wave packet such as a tsunami (Vadas et al., 2015). The small-amplitude Lamb waves in the MLT could be the source of the enhanced GW activity before the HTHH eastward GW actually arrived in the observation volume (Stober et al., 2023). Similar observations can be made for the MEN and KUN meteor radars in China, which also reveal an increased variability before the westward HTHH GWs arrived (see Figure 7, and 6).

The observed phase velocities derived in the present study depend crucially on how the timing is determined and referenced. In the literature, typically the onset of the HTHH eruption sequence is used as a reference (15th January 04:15 UTC). However, this captures only partially the complexity of the eruption sequence, which lasted for about three hours and consisted of several vigorous explosions. We accounted for this complexity by always representing the HTHH GW packet by vertical lines where

the left one refers to the onset and the right one is given by onset +3 hours. This is important for the comparison with HIAMCM where 5 primary volcanic eruptions are modeled with different launch times spread over 1 hour 30 min after the initial primary explosion (Vadas et al., 2023a). Furthermore, there is a longitudinal and meridional shift of the location of the updrafts from

GOES-17 of about 0.7° or about 70-90 km, and the nominal geographic coordinates of the volcano (175.38° W, 20.54° S), which we used as a reference to compute the distances. Considering these aspects demonstrates how precisely the $t_0$-time and also the location of the eruption have been inferred from the global meteor radar data.

A key factor for the propagation of the GWs is the speed of sound because it forms an effective natural altitude-dependent bottleneck or barrier. Since most of the large-amplitude secondary GWs were created at z$\simeq$120-150 km (e.g., Fig. 13 of Vadas

et al., 2023b), those that propagate through the MLT are downward-propagating. The mesopause region typically has a sound speed of $c_s \simeq$280 m/s, which results in a maximum intrinsic horizontal phase speed of $c_s \simeq$250 m/s (Vadas and Azeem, 2021). This is consistent with the maximum observed phase speeds of 240 m/s of the HTHH GWs in the MLT. In addition, the eruption occurred during the southern hemispheric summer, where temperatures below 140 K are expected, which was confirmed by the presence of Noctilucent clouds in Tierra del Fuego. These temperatures for the lower southern latitudes around the Antarctic

continent are consistent with a speed of sound limit close to approximately 239 m/s (Stober et al., 2023), which results in a maximum intrinsic horizontal phase speed of $0.9c_s \simeq$215 m/s. This threshold is very likely the reason why the MLT winds at DAV and McM did not show a large amplitude or clear wave pattern of the HTHH GWs at both sites, although both stations are located at the closest distance to HTHH of all meteor radars included in our study. However, the HIAMCM predicted that McM and DAV would observe HTHH GWs with observed phase speeds of about 240 m/s. As mentioned previously HTHH

GW packet was Doppler shifted due to a strong Q2DW activity in the southern hemisphere (Stober et al., 2023), which was also found in ROT, TDF, McM, and DAV (data not shown) and, thus, explains the differences in the arrival times between the observations and HIAMCM. However, it also confirms that the intrinsic wave speed was much closer to the sound speed barrier and, thus, forced the HTHH GWs to reflect, thereby preventing the GWs from entering the Antarctic polar MLT region. Hence, the HTHH GWs signatures in the MLT winds found above McM and DAV appear to be much weaker.

Comparing the observations from MEN, KUN, ALO, and CAR confirms findings presented in a previous study about the intrinsic wave properties of the HTHH GW traveling along the GC (Stober et al., 2023). All four stations are located rather close to the same GC and, thus, are suitable to evaluate the results obtained from the high-resolution 3DVAR+DIV retrievals (Stober et al., 2023). Hence, our present study confirms that the HTHH GW packet had an intrinsic wave speed of 202.5 m/s along this GC. This velocity is low enough to not be affected by the speed of sound limitation caused by a too-cold mesopause

somewhere along the GC, which is confined between 40° S and 40° N. This might also explain why only the 3DVAR+DIV analysis at CONDOR revealed the signatures of both the eastward and westward HTHH GW packets.

Simulations of the Lamb wave modes L0 and L1 with WACCM-X suggest that the L1 has a phase speed of about 240 m/s (Liu et al., 2023). Although this phase speed seems to agree with the observed phase speeds in this study, the observed horizontal wavelength of about 1600-2000 km (Stober et al., 2023), and a wave period of 2 hours 20 min cannot be reconciled with the

Lamb waves properties. Furthermore, HIAMCM and meteor radar zonal and meridional winds exhibit variable phase relations between both wind components and a much less coherent phase front than expected for a Lamb wave. There might have been

an L1 mode in the MLT, but such a wave was hardly visible in the available meteor radar wind data. Also, note in this context that the spatial averaging of 350 km in diameter and temporal resolution of 10 min clearly sets limits for the observed radar winds to reflect Lamb wave perturbations.


## 9   Conclusions

In this study, we analyzed winds from 13 globally distributed meteor radars to track HTHH GW packets in the eastward and westward directions from the volcano. The observational data was compared to HIAMCM simulations. This allowed us to identify the HTHH GWs in the observations and applied the same observational filter to the HIAMCM. We were able to
track eastward propagating HTHH GWs for 30000 km from the eruption site in the Pacific up to the Arctic over Svalbard. We determined an observed horizontal phase velocity of 240±5.7 m/s. Furthermore, we found westward propagating HTHH GWs with an observed horizontal phase speed of 166.5 ±6.4 m/s. The observed phase speeds are in excellent agreement with HIAMCM, which showed observed phase speeds of 239.5±4.3 m/s and 162.2±6.1 m/s for the eastward and westward GWs, respectively.

The joint analysis of the global meteor radar observations and HIAMCM perturbations demonstrate the interplay between data and simulations, with both informing each other helping to understand the complex dynamical situation caused by the HTHH eruption. The model data provided global information about the HTHH GW propagation, which confirmed that the European meteor radars observed the eastward GW as the strongest signature of the HTHH eruption at the MLT. On the other hand, the observations also revealed that the Q2DW played an important role in the propagation along different GC. Furthermore, the
MR observations alone are not sufficient to distinguish between primary and secondary GW generated by HTHH. However, the modeled HIAMCM secondary waves explain very well the observed phase speed and propagation direction and, thus, there is some confidence that indeed these GWs were forced due to the dissipation of primary waves.

The comparison between the meteor radar winds and corresponding model results illustrates the capability of HIAMCM to model the HTHH GWs and their propagation over long distances. The HTHH eruption provided an ideal benchmark for
evaluating and comparing observations and models regarding the global dynamics of GWs from a spatially and temporally confined source region. The HIAMCM result revealed the very complex structure of the wavefront generated by the HTHH eruption in the MLT with its large variations of the background flow for GWs from tides and traveling PWs as opposed to the situation observed in the stratosphere (Wright et al., 2022).

Tracking the eastward HTHH GW in the meteor radar winds made it possible to determine the GW generation time to be
within 6 min of the nominal eruption time (15th Jan 2022 04:15 UTC) and the location was found to be within a range of 78 km to the WGS84 coordinates of the volcano. Both values are in good agreement with HIAMCM and within derived statistical uncertainties. Furthermore, the obtained accuracy of the determination of the eruption location from HIAMCM is quite satisfactory when considering the complexity of the eruption sequence modeled with MESORAC, which is based on GOES-17 updrafts. These updrafts extend over 0.7° in longitude and showed a horizontal distance of up to 70 km from the

volcano WGS84 coordinates. Only the secondary GWs that resulted from the ambient flow effects that occurred when the updraft-induced primary GWs dissipated were simulated by the HIAMCM and, hence, these secondary GWs (not the primary GWs) propagated around the globe and can explain the majority of the analyzed worldwide radar wind GW perturbations in response to the HTHH eruption.

*Author contributions.* The meteor radar data analysis was performed by GS. HIAMCM runs were carried out by SLV and EB. The data interpretation was done by GS, SLV, EB, JC, DM, SN, and KB. All authors contributed to the editing of the manuscript. The operation of the meteor radars was secured by all instrument PIs.

*Competing interests.* All authors declare that there are no competing interests.

*Disclaimer.* Any opinions, findings, and conclusions or recommendations expressed in this material are those of the author(s) and do not necessarily reflect the views of the National Science Foundation.

*Data availability.* HIAMCM wind fields can be requested from Sharon Vadas (vasha@nwra.com). The meteor radar data can be requested from the instrument PI's for Dav(Damian.Murphy@aad.gov.au), TDF(diego.janches@nasa.gov), ROT(tmof@bas.ac.uk), SVA and NORDIC (njal.gulbrandsen@uit.no,tutumi@nipr.ac.jp,alexander.kozlovsky@oulu.fi, Kero@irf.se), McM(scott.palo@colorado.edu), ALO(liuz2@erau.edu), Pokerflat (dlthorsen@alaska.edu), CMO (pbrown@uwo.ca) and for CAR (vania.andrioli@inpe.br, paulo.batista@inpe.br, rburiti@df.ufcg.edu.br). The Mengcheng and Kunming radar data was provided through Wen Yi (yiwen@ustc.edu.cn). The retrieved 10-minute winds can be requested from Univeristy Bern (gunter.stober@unibe.ch).

*Acknowledgements.* Gunter Stober, Witali Krochin, and Guochun Shi are members of the Oeschger Center for Climate Change Research (OCCR). Witali Krochin and Guochun Shi are supported by the Schweizerischer Nationalfonds zur Förderung der Wissenschaftlichen Forschung (grant no. 200021-200517 / 1).

SLV and EB were supported by NSF grant # AGS-1832988.

Alan Liu is the CONDOR instrument PI and his work is supported by (while serving at) the National Science Foundation (NSF), USA. Zishun Qiao and the operation of the CONDOR meteor radar system are supported by the NSF grant AGS-1828589.

The Esrange meteor radar operation, maintenance, and data collection were provided by the Esrange Space Center of the Swedish Space Corporation.

This research has been supported by the STFC (grant no. ST/W00089X/1 to Mark Lester).

This study is partly supported by Grants-in-Aid for Scientific Research (no. 21H04516, 21H04518, 21H01144, 20K20940) of the Japan Society for the Promotion of Science (JSPS).

Federal University of Paraíba (UFPB) for facilities to install and operate the Cariri meteor radar.

Njål Gulbrandsen acknowledges the support of the Leibniz Institute of Atmospheric Physics (IAP), Kühlungsborn, Germany for their contributions to the upgrade of the TRO meteor radar.

425 Operation of the Davis meteor radar was supported by Australian Antarctic Science projects 4445 and 4637. Support for DJ as well as SAAMER-OS' operation are provided by NASA's Planetary Science Division Research Program, through ISFM work packet Exospheres, Ionospheres, Magnetospheres Modeling at Goddard Space Flight Center and NASA Engineering Saftey Center (NESC) assessment TI-17-01204. This work was supported in part by the NASA Meteoroid Environment Office under cooperative agreement no. 80NSSC21M0073.

The work by Wen Yi is supported by the National Natural Science Foundation of China (grants No. 42174183).

430 Poker Flat Meteor Radar is supported by NSF grant (AGS-1651464).

Christoph Jacobi acknowledges support by Deutsche Forschungsgemeinschaft, grant No. JA 836/47-1.

This research was supported by the International Space Science Institute (ISSI) in Bern, through ISSI International Team project 23-580 - Meteors and Phenomena at the Boundary between Earth's Atmosphere and Outer Space.

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

## Appendix A: Comparison of HIAMCM results and meteor radar observations

The appendix Figures compare the meteor radar observations and HIAMCM wind perturbations at a much higher temporal
resolution. The data is always centered around the theoretical arrival time for the eastward/westward HTHH GW inferred from the observed phase velocity fit shown in Figure 8.

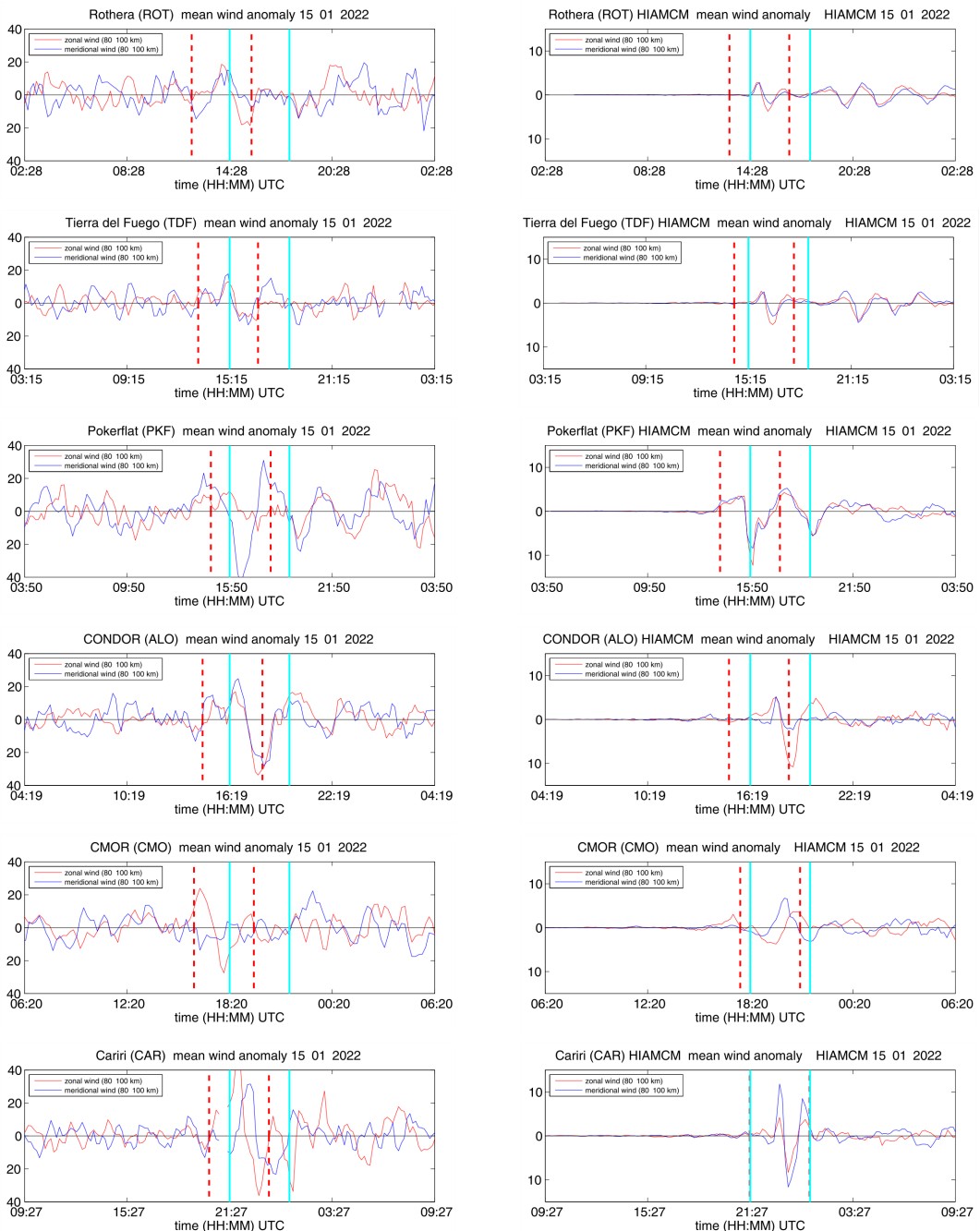

**Figure A1.** Wind anomalies centered at the arrival time for the meteor radars located eastward of HTHH and corresponding anomalies from HIAMCM. The cyan vertical lines span the predicted arrival of the eastward propagating GW. The black vertical lines indicate the timing for the westward propagating HTHH GW packet. The dashed vertical lines indicate the one-sigma interval of the time picks for each station. Note that the HIAMCM shows perturbations only to the right of the yellow line because we plotted the differences between the Tonga run and the base run.

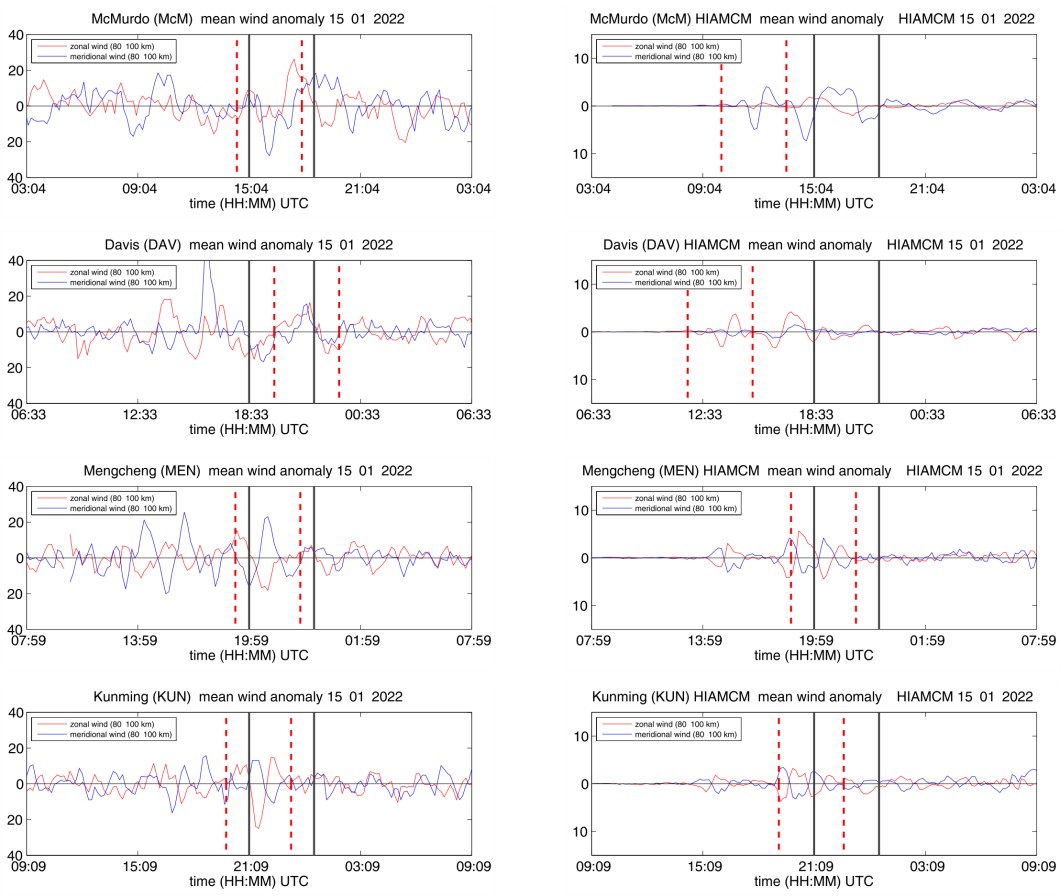

**Figure A2.** The same as Figure A1, but for the stations westward of HTHH outside the European continent.

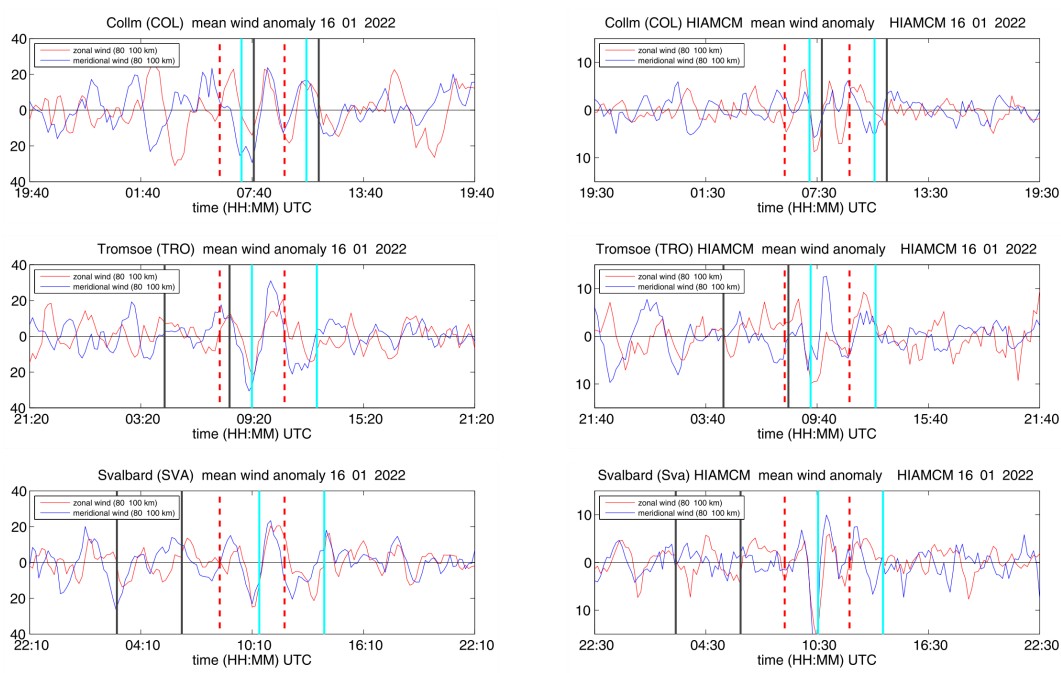

**Figure A3.** The same as Figure A1, but for the European meteor radar sites.