# Peer review of "Gravity waves generated by the Hunga Tonga-Hunga Ha‘apai volcanic eruption and their global propagation in the mesosphere/lower thermosphere observed by meteor radars and modeled with the High-Altitude General Mechanistic Circulation Model"

_EGUsphere, 2023_

## Author Response (AR1)

Reply to reviewer #2:

We thank the reviewer for the comments and suggestions. We will include additional figures and expand the recommended discussion of the importance of the Q2DW. We remove all statements about the HIAMCM referring to the polar vortex, as this is not the main narrative of the paper. However, we include in the reply some additional data from MERRA2 to show the location of the polar vortex. We added some additional paragraphs about the polarization relation of the HTHH GW and the importance of the Q2DW. Furthermore, the model session was expanded and includes more details and additional analysis.

We provide a track changes version prepared with latexdiff.

**General comments**

**Comment: Experimental setup**

A more detailed (but concise) description of the modeling setup (as described in Vadas et al. 2023a) should be included, which includes a discussion of the impact of dissipation and resolution on the analysis presented in the current work.

**Reply:**

We will expand this section as suggested, although the main narrative of the paper is the data analysis of the meteor radars and the comparison of observations to the model results and a dedicated modeling paper might be more useful to shed light on these aspects of the HIAMCM model.

**Comment:**

By the nature of the experimental design, only the effects of secondary gravity waves can be investigated using the HIAMCM. In the discussion (lines 277-278), it is mentioned that the HIAMCM zonal and meridional winds exhibit variable phase relations between both wind components and a much less coherent phase front than expected for a Lamb wave. However, there is no mention of Lamb waves being present in the analysis or HIAMCM results. Can the authors elaborate on how the results support this discussion point?

**Reply:**

This statement refers to the WACCM-X modeling of the Lamb wave presented in Liu et al., 2023 (Figure 1). Although, the Lamb wave is not the main focus of this paper, it is worth to point out the main differences between the different observations and modeling performed with a number of models. The Lamb waves propagate only at 2 different speeds in Liu et al, while the secondary GWs propagate at a variety of speeds from ~100 to 750 m/s (Vadas et al, 2023b, c).

**Comment:**

A discussion should be included on the fact that the HIAMCM model generally underestimates GW amplitudes by ~50% and up to 150% (Vadas et al., 2023a), which would then include a comparison between the fitted observed and simulated GW packet amplitudes. Currently, it is difficult to make out whether or not the HIAMCM model adequately captures the wave forms needed to explain the observed HTHH wave-response. The concluding statement (lines 312-315) that the secondary GWs can explain the majority of the analyzed radar wind GW perturbations therefore does not carry much weight, given also that other possible sources of wind perturbations are not discussed in detail.

**Reply:**

The biggest challenge comparing the HIAMCM model results, and the observations is caused by the way on how the HTHH GW perturbations are extracted, which also has an impact on the derived amplitudes. The meteor radar data can only be filtered with a temporal bandpass including also scales that are not resolved in the model. Furthermore, the mean and large-scale flow of the reference run depends on the nudged data set, which is MERRA2. Unfortunately, MERRA2 does not capture some large-scale feature such as the Q2DW. All deviations of the large-scale flow in MERRA2 will also manifest in HIAMCM and, thus, a one to one agreement is not expected.

**Comment: Polar vortex and Figure 2**

The statement that HIAMCM shows that the polar vortex disturbs the westward traveling GWs (lines 120-124) should be discussed in more detail. At first glance, a discussion of the impact of the polar vortex on the HTHH GWs is not given in the cited literature (e.g. Vadas et al., 2023b). For example, what are the mechanisms behind the polar vortex altering GW propagation, and what was the state of the polar vortex during the HTHH event? Fig. 2 does not seem to show any clearly discernible polar vortex structure, as alluded to in the text.

**Reply:**

We will reword the statements about the polar vortex concerning HIAMCM as a detailed analysis of these GWs is beyond the scope of the paper. However, the situation above Europe was rather similar to the case study presented in Vadas et al., 2023a, investigating a case study from 12-14 January 2016, which was also already analyzed in detail in Stober et al., 2017 and Matthias and Dörnbrack, 2018.

We have added a summary Figure below about the polar vortex structure during the eruption in the northern hemisphere using MERRA2. HIAMCM is nudged to MERRA2 and, thus, reflects the same polar vortex structure. In this study, we only use data between 80-100 km. The Figure below shows MERRA2 zonal and meridional winds, temperature and ozone at 71 km. The zonal winds reach values of about 100 m/s on the northern hemisphere.

[Figure]

MERRA2 15-Jan-2022 06:00:00 altitude: 70 km

**Comment:**

On lines 175-176 the impact of the polar vortex on the observed wind variations is further noted as being caused by GWs generated by the vortex. As a reader, it is impossible to verify such a statement, given also the other possible sources of wind variability mentioned in the text. While it is well known that the polar vortex can generate GWs, how and to what extent that applies to the current analysis should be discussed in more detail (including a discussion on the current state of the vortex). It is further noted that orographic GWs add noise to the European sector winds. But can orographic GWs not just as well add noise to the South American sector winds?

**Reply:**

The statement refers to the increased activity in the filtered meteor radar winds at Collm and Tromso and to some extend at Svalbard. Orographic waves can contribute to the total variability and thus will be mentioned in the revision as well. However, we are confident that the polar vortex plays a major role for the remaining variability. This becomes clearer when comparing to Davies, McMurdo, Tierra del Fuego and Rothera winds shown in Figure 3. Meteor radars inside the polar vortex show an increased variability compared to meteor radars in the southern hemisphere even at the same conjugate latitude. However, adding orographic waves as additional source won't change the conclusions. We will revise the HIAMCM section on GW over the European sector.

**Comment:**

Could the authors discuss the source of wind perturbations above Europe and Scandinavia in Figure 2? These are already present at the 12:00 snapshot (Fig. 2a), and grow rapidly before the GWs reach the sites (16:00 snapshot, Fig. 2b), with the 'noise' quickly becoming comparable in magnitude (or even greater than) the GW signal itself. It seems hard to argue

that this is due to the polar vortex, since both the 'Tonga run' and 'base run' simulations specify the same vortex structure, and the noise is limited to the European sector.

**Reply:**

The core of the polar vortex is well-known to generate/amplify GWs (Becker et al, 2022b). In this scenario, the GWs extract energy from the wind shear of the polar vortex. A recent study was published where the polar vortex was disrupted by a planetary wave event during 12-14 January 2016, causing a core fragment of the polar vortex to lie over Europe and generate a plethora of GWs over Europe (Vadas et al, 2023a). That paper showed that the GWs from the polar vortex are generated by the wind shear on the outer edge/bottom side of the polar vortex. Rayleigh lidar observations over northern Norway showed secondary GWs that were generated by the dissipation of the primary GWs from the polar vortex, which agreed well with the results of the HIAMCM. These secondary GWs were generated in a similar manner as here in the Tonga case, namely the rapid deposition of momentum from the dissipation of primary GWs destabilizes the atmosphere, which responds by radiating secondary GWs. That study also showed that the primary GWs from the polar vortex were observed by AIRS (temperature perturbations) and that those temperature perturbations agreed well with those modeled by the HIAMCM, thus validating the HIAMCM as a useful model to study the generation mechanism of these GWs. Interesting, the polar vortex on the day of Tonga seems to also be disturbed by a planetary wave event (2 main fragments, one over Europe—see above), with a similar morphology as what occurred during 12-14 January 2016 (Vadas et al, 2023a). Due to slight differences in the runs, some GWs are not entirely removed when subtracting the HIAMCM Tong run from the no-Tonga (base) run.

**Figures 3, 4 & 5**

**Comment:**

These figures should have labels added to the axis, and the font size should be enlarged to make the legend and labels more readable. Due to the small panel size, variations in the observed winds are difficult to distinguish, giving the figures an overall noisy impression. The y-axis scale on the observation can probably be changed to limits [-40, 40] without loss of information.

**Reply:**

We will adapt the amplitude scale as suggested. The amplitude scale was selected to ensure the comparability to the first HTHH paper using the spatially resolved winds from CONDOR and the Nordic Meteor Radar Cluster. The 3DVAR+DIV permits to reduce the spatial averaging intrinsic to the observational filter of the monostatic analysis (300 km diameter) compared to 30 km spatial resolution with the tomographic analysis.

**Comment:**

Throughout the text it is argued that increases in wind variability can be identified by looking at these figures (e.g., lines 176-178 and line 239), but this is not so straightforward. Perhaps increases or decreases in wind variability at certain time periods can instead be quantified, for

example using a sliding window approach of the calculated variance. Currently, it is difficult to judge if a supposed increase in variability (e.g. before the arrival of the westward HTHH GWs at the MEN radar) is just that, or if the increased in variability is due to the presence of some (possibly unaccounted for) waveform.

**Reply:**

The radar data certainly captures also other wave sources including the unresolved scales compared to HIAMCM. (The HIAMCM can only resolve GWs with horizontal wavelengths >156 km). In particular, for all stations that are considered westward of the eruption side the HTHH GW amplitude is not much larger than other sources during January outside the shown data. We identified the HTHH wave by the coherence and phase relation between the meridional and zonal wind component. A statistically significant increase of the variability is only found for some stations. However, it is still possible to determine the arrival time of the HTHH signature to estimate the phase velocities and to determine the eruption time and source region. The second clear indicator of the HTHH GW is given by the vertical wavelength. Below there are two examples from Collm and Kunming to visualize the signature in the vertically resolved zonal and meridional winds. The analysis procedure outlined below was applied to all stations and repeated with different retrieval setups concerning the temporal resolution.

Below we show altitude resolved zonal and meridional winds and the corresponding residual winds after applying our temporal filter. Inside the black oval both radar winds show a clear distortion caused by a GW with a very long vertical wavelength. A similar feature can be found in Kunming.

[Figure]

The residual winds are shown below. The HTHH GW is indicated inside the black oval as GW with a very long vertical wavelength. The Collm radar exhibits how exceptional the HTHH GW is compared to the typical GW spectrum that is found before the eruption.

[Figure]

**Comment:**

As the left-hand and right-hand panels of Fig.3,4,5 show fundamentally different things, it would be beneficial if the fitted waveforms and amplitudes are overlain (and possibly zoomed in on) for both the observatory and simulated winds. In addition, showing the observed and modeled wind fluctuations above and below each other (rather than side-by-side), would make the difference in arrival time easier to distinguish.

**Reply:**

We will add a zoom in version of the HIAMCM and meteor radar winds. Below we added some examples for selected stations to visualize the agreement between the model perturbations and the observations. However, there is no reason to expect a one-to-one agreement. Subtracting the perturbation run from a base run is not identical to the spectral filtering and vertical integration of the meteor radar winds. Thus, differences between the amplitudes and some wave morphology is expected.

Westward wave:

[Figure]

[Figure]

Eastward wave:

[Figure]

**Comment:**

As mentioned earlier, HIAMCM simulated GWs are smaller by ~50% than observed due to the choice of turbulent diffusion coefficient D0 (Vadas et al. (2023a)). The impact of D0 on simulated GW amplitudes should therefore also be discussed within the context of Fig.3,4,5, including how it impacts the choice of y-axis scaling for the right-hand plots, and the amplitude of the fitted waveforms.

**Reply:**

The main focus of the paper is the data analysis of the meteor radars and HIAMCM wind fields rather than how these are obtained. However, we added a statement referring to Vadas et al., 2023b : "D0 is tuned to 2,000 m 2/s because it results in GW amplitudes that are closest to the ICON-MIGHTI amplitudes without destabilizing the HIAMCM (see Section 5); larger values of D0 result in smaller GW amplitudes, which is inconsistent with the ICON data."

Furthermore, we want to emphasize that so far, no other model was compared with such a level of detail to the mesospheric observations or was able to reproduce the observed GW amplitudes.

**Comment Q2DW:**

The impact of the Q2DW on the GW propagation lacks detail. It is not clear from the text or earlier cited work what the amplitude of the Q2DW is and how it compares to the concurrent tidal and low-frequency GW amplitudes. Discussing the mechanisms of the Q2DW impacts is important, as one of the concluding statements of this work is that the observations reveal that the Q2DW played an important role in the GW propagation.

L263 - Stober et al. (2023) are cited in support of the statement that the GW packet is doppler shifted due to a strong Q2DW, thus explaining the difference in arrival time between the observations and HIAMCM for McM and DAV. It is unclear how the cited work supports the current statement; could this be elaborated further?

L205-208 That the GW packet faced a strong headwind towards the south due to the presence of the Q2DW would surely depend on the phase of the Q2DW in the region of propagation? While on lines 204-205 it is clarified that the Q2DW meridional wind component showed clear northward winds over the South American continent, the Antarctic stations (and the associated propagation paths) are at considerably different longitudes (as shown in Fig. 1). Given the zonal wavenumber 3 structure of the Q2DW, it is not obvious what the phase of the Q2DW was (i.e., headwind or tailwind) facing the Antarctic stations, and it may well have been a tailwind.

While the Q2DW was said to be present, its magnitude and relative importance versus the semidiurnal and diurnal tidal mode amplitudes is not discussed. That the superposition of these wave components leads to high wind speeds, does not necessarily imply that Q2DW winds are especially large (even though this may very well be the case), or that the semidiurnal and diurnal tides do not have a similarly large impact on GW propagation. The presence of other (large amplitude) tidal waves along the different great circle propagation paths (and their representation in the HIAMCM model), should therefore be discussed in more detail.

**Reply:**

We analyzed the Q2DW from the available meteor radar data. A scheme of the spatial structure of the Q2DW as derived from the radars is shown below for two separate times.

As shown in the Figures below the HTHH GW that travels towards the Antarctic stations at McMurdo and Davis propagated against a strong head on wind caused by a Q2DW with wavenumber 3. The phase behavior of the Q2DW was determined analyzing the low pass filtered mean winds from Rothera, Davis, SAAMER (Tierra Del Fuego), CONDOR and Cariri. These winds were filtered with the adaptive spectral filter (ASF2D, Stober et al., 2020, Baumgarten and Stober, 2019 ). The best match was achieved for a wavenumber 3. We also tested a wavenumber 2, but it was not possible to get the observations from Davis and SAAMER as well as CONDOR to a similar agreement. The period of the Q2DW can be either directly inferred from the zonal and meridional winds below or through a wavelet spectrum. The amplitude is rather different between different geographic locations, and we estimated about 60 m/s as maximum around 30°S. Considering the longitudinal difference of

about 140° and the out of phase behavior between Davis (77.969354°E) and Rothera (68.119713°W) it is possible to estimated/infer the wavenumber structure of the Q2DW.

The Figure below shows the structure of the Q2DW at two different times as inferred from the meteor radars for the meridional wind component. Our analysis indicates that the HTHH GW that propagated towards Antarctica along the great circles connecting McMurdo and Davis did encounter prevailing winds with a strong northward component reaching about 60 m/s around 30°S (estimated from CONDOR).

[Figure]

[Figure]

Hunga Tonga-Hunga Haʻapai 15-Jan-2022 11:15:00

The Figures below show the daily mean zonal and meridional mean winds estimated by the ASF2D for all southern hemispheric radar stations, which permits to remove the tidal and all low frequency waves. The Q2DW can be identified in the zonal and meridional winds for some stations. We used the information collected from all of the meteor radars shown below to determine the spatial structure of the Q2DW.

[Figure]

**Other comments**

**Comment:**

L17 - The wording would be more scientific if 'huge' is removed.

**Reply:**

Done.

**Comment:**

L22 – Total Electron Content (TEC) abbreviation should be defined.

**Reply:**

Added.

**Comment:**

L43 – The way the sentence starting with "In this study,..." is worded, suggests that TEC perturbations are also discussed in the current work, which they are not.

**Reply:**

Reworded.

**Comment:**

L47 - "strongest gravity waves launched by the HTHH". Maybe this is a matter of semantics, but can secondary GWs really be considered as being launched by the HTHH? Perhaps 'resulting from' would be more appropriate.

**Reply:**

Changed.

**Comment:**

L62 – If the 2 km resolution with a 5-kilometer vertical averaging window centered around the respective altitude is identical to the technique described in Stober et al. 2023, this article should be cited here. If the methodology has been substantially modified since Stober (2023), more information about the analysis technique should be given instead.

**Reply:**

Added. We applied the same retrieval setup.

**Comment:**

Fig. 2 - The quality would be improved if the axis labels would not overlap with the plot.

**Reply:**

Changed.

**Comment:**

L77 - "Could" to "can".

**Reply:**

Done.

**Comment:**

L78 – The dashed red line not only connects (or lies close to the) ALO and HTHH, but also KUN, MEN, and CAR, as noted in the caption of Fig. 1. The caption and text statements should be the same.

**Reply:**

Changed.

**Comment:**

L92 – Sentence can be clarified by rewording to "...specify realistic large-scale meteorological fields through which the resolved GWs propagate (Becker et al., 2022)".

**Reply:**

Done.

**Comment:**

L100 - "cover" to "covering"

**Reply:**

Done.

**Comment:**

L124 - 'This is consistent…' should be reworded to clarify what exactly 'this' refers to. Currently it can be read as if the cited work also supports the result that the polar vortex disturbs GW propagation, which does not appear to be the case.

**Reply:**

The statement about the polar vortex is based on an analysis of MERRA2 data (see above). The polar vortex on the northern hemisphere was well-established at the day of the eruption.

MERRA2 winds indicate a presence of the polar vortex from the stratosphere to the mesosphere. The HTHH GW had to cross or travel against the polar vortex eastward wind jet depending on the geographic position of the meteor radar and, thus, the GC path. All GC for the meteor radars in the European sector have GC paths crossing the polar vortex. However, we will reword the sentence to avoid to strong statements in that regard. Furthermore, we want to emphasis that MERRA2 did not indicate a signature of the Q2DW in the southern hemisphere at 71 km altitude.

**Comment:**

L144 - "theoretical arrival times" suggests the arrival times are based on theory, while they are derived from best-fit phase speed using all stations that detected this wave. Perhaps 'derived' would be more appropriate?

**Reply:**

The term theoretical shows the arrival times of the best fit for all stations. Thus, we changed the wording as suggested.

**Comment:**

L145-150 In the discussion (L243 - 244), it is mentioned that the distance between the solid left and right vertical lines is 3 hours. Please add this information here already.

**Reply:**

Added.

**Comment:**

L152-154 Mentioning the power of the CMOR meteor radar relative to the other radars, suggests that the radar power may represent a considerable source of variability for the other radars. Considering also the impact of diurnal meteor count variations on wind variability alluded to on lines 163-165, the impact of the system power as well as diurnal meteor count variations on wind variability should be discussed in more detail.

**Reply:**

We added a sentence.

**Comment:**

L165 Perhaps 'This provides…' can be reworded to 'Our results therefore support…', as it currently reads as if the meteor count rate itself somehow impacts asymmetric azimuthal GW propagation.

**Reply:**

Done.

**Comment:**

L217 'The reasons for..' could be changed to 'Possible reasons for…', since the discussion of the Q2DW impact on the DAV and McM stations is largely speculative. Furthermore, the discussion on lines 254-266 mention the impact of Southern Hemisphere MLT temperatures, which does not make sense chronologically. Merging the Q2DW and MLT temperature impacts on the DAV and McM stations into a single section would make sense.

**Reply:**

Will be revised as suggested.

**Comment:**

L223 - MR should be defined.

**Reply:**

Done.

**Comment:**

L252-253 In Vadas and Azeem, the mesopause sound speed (at 98 km) was estimated to be 270 m/s and not 280 m/s. It would also be nice to see in the text for which conditions this number was obtained (March 25 at 100°W and 34°N).

**Reply:**

We will add more details.

**Comment:**

L270 - It is unclear where the wave speed of 202.5 m/s comes from. This number has not been mentioned earlier in the text.

**Reply:**

In Stober et al., 2023 we estimated the intrinsic phase speed of the HTHH GW from CONDOR 3DVAR+DIV retrievals to be 200-212 m/s. Thus, we found in this study a bit more precisely a number of 202 m/s, which falls between our first estimate. We will add more context in paper.

**Comment:**

L279 - It is unclear what the visibility of a Lamb wave in the meteor radar wind data entails (Lamb waves are noted as being "hardly visible" - can evidence of this wave somehow be seen in Fig 3,4,5?). The sentence starting on line 279 also seems to imply that Lamb waves are effectively filtered out by the spatial and temporal averaging kernel employed in this work. If so, the statement that the observed wave period and horizontal wavelength observed in this work (lines 275-277) cannot be reconciled with that of a Lamb wave does not make sense, since by design, Lamb waves cannot be measured.

**Reply:**

We identified the Lamb wave in Stober et al., 2023 Figure 6. We were not able to identify a similar structure in the standard monostatic analysis applied in this study. The retrieval used for the 3DVAR+DIV had a temporal resolution of 5 min. We will rephrase the sentence accordingly.

**Comment:**

Fig. 7 - It would be good to label the ALO stations for the eastward and westward wave packets separately, and to add gridlines to the figure.

**Reply:**

Done.

**Comment:**

Many of the DOI's in the reference list are formatted wrongly, having "https://doi.org" twice.

**Reply:**

We need to revise this format of the bibliography.

Reply to reviewer #3:

We thank the reviewer for his helpful and constructive feedback on our submitted manuscript. Below we provide a point-by-point reply for the discussion and will revise the paper accordingly. As suggested by the reviewer, we will expand the modelling section in the revision, although the main narrative of the paper are the data analysis and observations. We also want to point out that all statements related to the HTHH-GW and our analysis refers to altitudes above the stratosphere and we are aware that the situation in the troposphere and lower stratosphere had different wave propagation characteristics. The suggested reference will be added in the introduction and discussion.

We have prepared a tracked changes version with latexdiff.

**Main comments:**

**Comment:**

1) The reasons for the two-step modeling are not explained in the paper. Perhaps they can be found elsewhere, but they should be repeated here. In particular, why not include the primary waves directly in HIACCM, and let them dissipate and directly trigger secondary gravity waves ?

**Reply:**

The forcing of the secondary GW from MESORAC is based on the dissipation from the primary GW and was constraint by the GOES-17 observations. The primary GW launched by HTHH did not travel 'far' away from the eruption side. The term 'far' refers here to a few thousand kilometers. Some secondary waves in HIAMCM travelled several times around the globe.

**Comment:**

2) It is argued that the waves observed by the meteor radars in the MLT are secondary gravity waves, and not primary waves (in particular not the mode referred to as L1 by Liu et al., 2023 or Pekeris wave by Watanabe et al., 2022). The line of argument does not sound compelling to me, since the propagation speed of the waves examined is similar to that of the L1 mode, as mentioned by the authors.

**Reply:**

The main conclusion of the paper is that the GW, which are seen in the meteor radar data, agree with the model perturbations caused by the HTHH eruption. The strongest argument for the GW nature of the mesospheric signals and perturbation in HIAMCM are provided by the polarization relations of GW. Along the great circle passing through the ALO, MEN and KUN, we were able to detect differences in the observed phase velocity between eastward and the westward propagation relative to the eruption site. Furthermore, the polarization between zonal and meridional wind perturbations are different in the different azimuth directions suggesting a GW origin. One other aspect that seems to be problematic for the L0 and L1 modes at the mesosphere are related to the speed of sound. Due to the cold mesopause temperature

such fast propagating modes are quasi-supersonic and, thus, are not expected to reach large amplitudes. However, the long vertical wavelengths might help to reach this modes to penetrate into the thermosphere and again become more visible.

**Comment:**

One argument I see in favor of the authors' interpretation is that the amplitude of the L1 mode is expected to be larger West of Tonga (e.g., Fig S1 of Liu et al., 2023), while the observations (in agreement with HIAMCM, apparently) suggest the opposite. Another possibility is that L1 does not extend to the MLT. Perhaps the authors could elaborate on this in the discussion.

**Reply:**

We add a figure and an additional section about the polarization between the zonal and the meridional wave amplitude and their phase behavior. In a previous study (Stober et al., 2023), we were able to identify the L0 mode. However, there was only a very weak signature and so far we were not able to identify a certain fluctuation in the wind to match the L1 mode.

**Comment:**

3) Related to the previous comment, the information about GW amplitudes at various distances and in various propagation directions is missing from the discussion and could be useful in the argumentation. Even if there is a discrepancy with the observations regarding the absolute amplitude, there is valuable information in the simulation about the relative amplitude of the fluctuations between the stations.

**Reply:**

This is a valuable point and we will add a section discussing the amplitude variations in dependence of the propagation direction.

**Comment:**

4) Figures:

Figure 2: At early times, the wave fronts in HIAMCM are quite concentric. Hence, a keogram at 96 km in the eastward and westward directions would help identify the wave propagation speeds in HIAMCM.

**Reply:**

We are going to add such a keogram from HIAMCM.

**Comment:**

In Figures 3, 4 and 5, the labels should be enlarged. The information on the distance is already provided in Figure 1 and does not need to be repeated, however the azimuth is missing.

**Reply:**

The figures have been updated and revised accordingly.

**Specific comments**

**Comment:**

l 34: ExB-coupling: please write the formula in latex with $$, or expand.

**Reply:**

Latex corrected.

**Comment:**

l49: 'at various distances from the eruption site': you might add 'at various distances and azi-muths'

**Reply:**

added.

**Comment:**

l 52: 'characteristic amplitude variations at certain stations' do you mean 'characteristic ampli-tude variations **between** certain stations' or 'characteristic waveform' ?

**Reply:**

Changed.

**Comment:**

l 61: I suggest rephrasing the statement regarding vertical resolution, which at present is con-fusing. A 2 km vertical resolution suggests that you can detect perturbation with a 4-km verti-cal wavelength. This is definitely not the case using a 5-km moving average.

**Reply:**

We added information and more details.

**Comment:**

around l 60: it would be useful for the reader to know which components of the wind vector are measured by meteor radars

**Reply:**

Added.

**Comment:**

l65: 'collecting' -> within ?

**Reply:**

Rephrased.

**Comment:**

l 65-71: For legibility, I would recommend removing the list of radar sites from the main text, and refer to Table 1. If you adopt this suggestion, you should add the references for the instruments to Table 1.

**Reply:**

The added references are the main reason for keeping it in the text, but we might will comply with the suggestion and put everything in the table.

**Comment:**

l 81: Is it necessary to include the 90 km altitude ? The difference is 1.5 %

**Reply:**

Its easy to remove this error and, thus, minimize a potential bias.

**Comment:**

Table 1: it would be good to recall the location of the volcano in the caption

**Reply:**

Added.

**Comment:**

l 89: Some information is missing from the HIACCM model description: grid spacing, both vertical and horizontal, upper boundary condition, ... They can probably be found elsewhere, but should be repeated here.

**Reply:**

We will add the model specifics.

**Comment:**

l 89-101: a few details on the workflow of MESORAC and its initialization would be welcome here. From the acronym, I understand that the local GW forcing related to the eruption (which is inferred from geostationary satellite images) is propagated/dissipated using ray tracing equations, but this needs to be explicitly stated. Furthermore, some additional information on the model would be welcome (even if it can be found in the Vadas et al. 2023 paper), e.g. what is used as background flow for the ray tracing ?

**Reply:** We added the following paragraph in the revision.

The Model for gravity wavE SOurce, Ray trAcing and reConstruction (MESORAC) calculates the primary GWs created from localized (in space and time) vertical updrafts of air using the Fourier-Laplace analytical fully-compressible solutions (Vadas, 2013). These updrafts, which are mechanical displacements of stratospheric/mesospheric air, are identified from NOAA's Geostationary Operational Environmental Satellite (GOES) data. The atmosphere responds by radiating concentric rings of GWs (Vadas et al 2009, Vadas et al, 2012). MESORAC ray traces these GWs forward in time, including their phases, and reconstructs the primary GW field using the GW phases and the GW dissipative dispersion and polarization relations (Vadas & Fritts, 2005, Vadas & Fritts, 2009). The background atmosphere is taken from the HIAMCM simulation for 15 January 2022 without the Tonga eruption (base case) using scales with lambda_H > 2000 km. Wave dissipation is due to molecular diffusion and turbulent diffusion from saturation. The body forces and heatings created by the dissipation of primary GWs are calculated as functions of space and time as in Vadas (2013). These ambient-flow effects are then added to the momentum and thermodynamic equations of the HIAMCM to simulate the secondary GWs from the Tonga eruption (Vadas et al, 2023).

**Comment:**

l 115-118: I am missing something here, do you mean that wind filtering occurred in the stratosphere West of the volcano, and this did not trigger 2ndary GWs ? The main difference I see in Fig. 2 is between the eastward and westward sectors.

**Reply:**

We rephrased this section to:

"This asymmetry (of the secondary GWs) is caused by the north and southward orientation of the local body forces (i.e., horizontal accelerations) in the thermosphere that are generated by the localized deposition of momentum that occurred at z~120-180 km where the primary GWs from Tonga broke and dissipated (Vadas et al, 2023a). These primary GWs were mainly propagating meridionally when they dissipated and created the body forces because the background wind was strongly meridional in the thermosphere. It is well-known that a local horizontal body force generates an asymmetric GW distribution, with the maximum

amplitudes being in and against the body force direction, and with zero amplitudes perpendicular to the body force direction (Vadas et al., 2003; Vadas and Becker, 2018)."

**Comment:**

l 122: 'Results from the HIAMCM also show that the HTHH GWs traveling westward along the GC towards Europe are disturbed by the polar vortex in the Northern Hemisphere and, thus, almost no coherent wavefront arrives over Scandinavia, although this would have been the shortest distance to HTHH.' : the second part of the sentence is confusing. Do you mean that ' almost no coherent wavefront arrives over Scandinavia **from the North-West (or another direction)**'.

**Reply:**

This part will be rephrased. However, indeed the wave front that approached from the north into the European sector had a much smaller amplitude and was less clearly visible in the data.

**Comment:**

There are some differences between the simulations in the European sector which are not related to HTHH (they emerge before the waves have reached that region) and might be hiding a HTHH signature there. Are you confident that this is not happening ?

**Reply:**

The waves generated in the European sector and those likely caused by the polar vortex are less coherent between the European meteor radars covering almost 28° in latitude. The wave packet that we identified as HTHH GW has similar properties in all three shown time series from COL, TRO and SVA. In particular, the vertical wavelength provided a more unified signature in all three stations. We add also a section on this aspect of the data analysis.

**Comment:**

l 123-124: 'gained strength over the Atlantic Ocean': for waves emanating from a point source on the sphere, we expect a geometric strengthening effect with refocusing of the rays after the mid-point to the antipodes (see, for instance, Matoza et al., 2022; Amores et al., 2022). Are you referring to that effect ?

**Reply:**

Yes. We added the explanation citations. The amplification is mainly due to the ray focusing close to the antipode in North Africa.

**Comment:**

l 125: add 'comparison' to the section title ?

**Reply:**

Done.

**Comment:**

l 144: 'all meteor radar': only a subset is shown in Fig. 3

**Reply:**

Changed.

**Comment:**

l 150: I would end the sentence after 'in the model'

**Reply:**

Changed.

**Comment:**

l 165: 'This provides... ' would be better placed right after 'on the same GC' on line 163

**Reply:**

Changed.

**Comment:**

l 201: I would rephrase 'The Q2DW may play an important role'-> 'The Q2DW may explain this discrepancy'.

**Reply:**

 Changed.

**Comment:**

l 200-205: A figure comparing the low-frequency wind field and the Q2DW between HI-ACCM (or MERRA-2) and meteor radar observations, for instance in an appendix or supplement, would strengthen this point.

**Reply:**

The revised manuscript will be updated with a paragraph about the Q2DW using the meteor radar data (see reply reviewer #2). Unfortunately, MERRA2 does not indicate the presence of a Q2DW and, thus, the wave is also not found in HIAMCM.

**Comment:**

l 256: 'noctilucent clouds in Tierra del Fuego' a reference would be welcome here

**Reply:**

The nocitucent cloud activity was shown in several conferences, but not yet published and, thus, we are not aware of a good reference.

**Comment:**

l 276: Apart from the introduction, it is the first time in this paper that mention is made of the horizontal wavelength. How is it derived ? The argument about the period being inconsistent with L1 is not really convincing: as far as I see from Fig. 3 (for instance at CONDOR or Cariri), HIAMCM sometimes also underestimates the wave period, and it does strongly underestimate the wave amplitude.

**Reply:**

The horizontal wavelength was derived in Stober et al., 2023 using the Nordic Meteor Radar Cluster and CONDOR using a 3DVAR+DIV retrieval. Here we only use the numbers that were found in this paper. The wavelength of the L0 and L1 mode were obtained from Liu et al., 2023.

**Comment:**

A data availability statement is missing. See https://www.atmospheric-chemistry-and-physics.net/policies/data_policy.html

**Reply:**

Will be added.

**Comment Typos:**

l 105: The sentence 'The HTHH GWs are extracted by subtracting a reference run from the disturbance simulation.' seems redundant with the previous one and could be deleted.

l 261: missing comma after previously

In the reference section, please check the url, some have twice the doi.org prefix.

**Reply:**

Corrected.

---

## Author Response (AR2)

**General reply:**

We thank the reviewer for his kind assessment of the revised paper and corrected the points listed below.

Comment from Referee 2:

**Comment:**

Only one very minor technical correction. Lines 187-188 of the annotated manuscript note that in Figure 5 the scaling of the y-axis of the modeled perturbations has been changed from 4 to 2.8, while the included figure still shows the original factor of 4 scaling. The updated factor 2.8 scaling does apply to Figure A1, however.

**Reply:**

We uploaded the revised Figure.

**Comment:**

Comment related to the review of the previous manuscript version by Referee 3 and your response:

"The noctilucent cloud activity was shown in several conferences, but not yet published and, thus, we are not aware of a good reference.": you could cite the corresponding presentation(s)/abstracts if available

**Reply:**

The source of the NLC statement is related to the abstract. However, this is beyond what is citable in the Journal, as no dates or other specifics are mentioned.
https://www.space.irfu.se/workshops/LPMR-EISCAT/EISCAT_LPMR_abstracts_220810.pdf

**Comment:**

Please include a data availability statement.

**Reply:**

Added.

Comment:

lines 496 and 533 : please correct the url

Reply:

Both DOI links are working from the pdf.

Other typos

**Comment:**

l 64: Thikonov -> Tikhonov

l 120: $D0$ -> $D_0$

In figure 3, Eastward->Northward and Westward->Southward

**Reply:**

Corrected.